*In Vitro* **Exposure to Isoprene-Derived Secondary Organic Aerosol by Direct Deposition**
**and its Effects on *COX-2* and *IL-8* Gene Expression**
Maiko Arashiro[1], Ying-Hsuan Lin[1,6], Kenneth G. Sexton[1], Zhenfa Zhang[1], Ilona Jaspers[1-5],
Rebecca C. Fry[1,3], William G. Vizuete[1], Avram Gold[1], Jason D. Surratt[1,*]
[1] Department of Environmental Sciences and Engineering, Gillings School of Global Public
Health, University of North Carolina at Chapel Hill, Chapel Hill, NC 27599, USA
[2] Center for Environmental Medicine, Asthma, and Lung Biology, School of Medicine,
University of North Carolina at Chapel Hill, Chapel Hill, NC 27599, USA
[3] Curriculum in Toxicology, University of North Carolina at Chapel Hill, Chapel Hill, NC
27599, USA
[4] Department of Pediatrics, School of Medicine, University of North Carolina at Chapel Hill,
Chapel Hill, NC 27599, USA
[5] Department of Microbiology and Immunology, School of Medicine, University of North
Carolina at Chapel Hill, Chapel Hill, NC 27599, USA
[6] Michigan Society of Fellows, Department of Chemistry, University of Michigan, Ann Arbor,
MI 48109, USA
*To whom correspondence should be addressed:
Jason D. Surratt, Department of Environmental Sciences and Engineering, Gillings School of
Global Public Health, University of North Carolina at Chapel Hill, Chapel Hill, North Carolina
27599, USA. Tel: (919)-966-0470; Fax: (919)-966-7911; Email: surratt@unc.edu
For Submission To: Atmospheric Chemistry and Physics Discussions
Manuscript Information: Number of Figures – 5.

**Abstract**

Atmospheric oxidation of isoprene, the most abundant non-methane hydrocarbon emitted into Earth's atmosphere primarily from terrestrial vegetation, is now recognized as a major contributor to the global secondary organic aerosol (SOA) burden. Anthropogenic pollutants significantly enhance isoprene SOA formation through acid-catalyzed heterogeneous chemistry of epoxide products. Since isoprene SOA formation as a source of fine aerosol is a relatively recent discovery, research is lacking on evaluating its potential adverse effects on human health. The objective of this study was to examine the effect of isoprene-derived SOA on inflammation-associated gene expression in human lung cells using a direct deposition exposure method. We assessed altered expression of inflammation-related genes in human bronchial epithelial cells (BEAS-2B) exposed to isoprene-derived SOA generated in an outdoor chamber facility. Measurements of gene expression of known inflammatory biomarkers interleukin 8 (IL-8) and cyclooxygenase 2 (COX-2) in exposed cells, together with complementary chemical measurements, showed that a dose of 0.067 $\mu g\ cm^{-2}$ of SOA from isoprene photooxidation leads to statistically significant increases in *IL-8* and *COX-2* mRNA levels. Resuspension exposures using aerosol filter extracts corroborated these findings, supporting the conclusion that isoprene-derived SOA constituents induce the observed changes in mRNA levels. The present study is an attempt to examine the early biological responses of isoprene SOA exposure in human lung cells.

## 1. Introduction

Recent work has shown that isoprene (2-methyl-1,3-butadiene) is an important precursor of secondary organic aerosol (SOA), which has potential impacts on climate change and public health (Lin et al., 2013b; Rohr, 2013; Lin et al., 2016). Current understanding of isoprene SOA formation is based on laboratory studies showing that gas-phase photooxidation of isoprene generates key SOA precursors, including isomeric isoprene epoxydiols (IEPOX), methacrylic acid epoxide (MAE), hydroxymethyl-methyl-$\alpha$-lactone (HMML), and isoprene hydroxyhydroperoxides (ISOPOOH) (Paulot et al., 2009; Surratt et al., 2010; Lin et al., 2012; Lin et al., 2013b; Nguyen et al., 2015; Krechmer et al., 2015). The formation of SOA from these precursors is influenced by controllable anthropogenic emissions such as oxides of nitrogen ($NO_x$) and sulfur dioxide ($SO_2$). Atmospheric oxidation of $SO_2$ contributes to particle acidity, which enhances isoprene SOA formation through acid-catalyzed reactive uptake and multiphase chemistry of IEPOX and MAE (Surratt et al., 2007; Surratt et al., 2010; Lin et al., 2012; Gaston et al., 2014; Riedel et al., 2015), while $NO_x$ determines whether the oxidation pathway leading to IEPOX or MAE/HMML predominates (Lin et al., 2013b; Surratt et al., 2010; Nguyen et al., 2015). Isoprene SOA comprises a large portion of global atmospheric fine particles ($PM_{2.5}$, aerosol with aerodynamic diameters $\leq 2.5$ $\mu$m) (Carlton et al., 2009; Henze et al., 2008) but few studies have focused on its health implications (Lin et al., 2016). Evaluating the health effects of SOA from isoprene oxidation is important from a public health perspective, not only because of its atmospheric abundance, but also because the anthropogenic contribution is the only component amenable to control (Pye et al., 2013; Gaston et al., 2014; Xu et al., 2015; Riedel et al., 2015).

Many studies have shown that particulate matter is closely linked to health effects
ranging from exacerbation of asthma symptoms to mortality associated with lung cancer and
cardiopulmonary disease (Dockery et al., 1993; Schwartz et al., 1993; Samet et al., 2000). $PM_{2.5}$,
in particular, has been linked to negative health outcomes with an estimated contribution of 3.2
million premature deaths worldwide as reported in the Global Burden of Disease Study 2010
(Lim et al., 2012). Despite evidence that particle composition affects toxicity, fewer studies
focus on the link between chemical composition and health/biological outcomes (Kelly and
Fussell, 2012). Prior work on complex air mixtures has shown that gaseous volatile organic
compounds (VOCs) alter the composition and ultimately the toxicity of particles (Ebersviller et
al., 2012a, b). SOA resulting from natural and anthropogenic gaseous precursors, such as α-
pinene and 1,3,5-trimethylbenzene, have been shown to affect cellular function (Gaschen et al.,
2010; Jang et al., 2006) and recently isoprene-SOA formed from the reactive uptake of epoxides
has been shown to induce the expression of oxidative stress genes (Lin et al., 2016).
The objective of this study is to generate atmospherically relevant isoprene-derived SOA
and examine its toxicity through *in vitro* exposures using a direct deposition device. Compared to
exposure of cells in culture media to resuspended particles, direct particle deposition likely
provides a more biologically relevant exposure model and enhances sensitivity of cells to air
pollution particle exposures (Volckens et al., 2009; Lichtveld et al., 2012; Hawley et al., 2014a;
Hawley et al., 2014b; Zavala et al., 2014; Hawley and Volckens, 2013). The Electrostatic
Aerosol *in vitro* Exposure System (EAVES) used in this study deposits particles generated in our
outdoor photochemical chamber directly onto lung cells by electrostatic precipitation (de Bruijne
et al., 2009). Similar techniques and devices have been used to expose cells to diesel exhaust
particles (Lichtveld et al., 2012; Hawley et al., 2014b), but our study is the first to utilize the
EAVES to explore the potential adverse effects of isoprene SOA on human lung cells.
Additionally, for a more atmospherically relevant exposure, isoprene-SOA was photochemically
generated in an outdoor chamber to mimic its formation in the atmosphere.
We have recently demonstrated through a chemical assay that isoprene-derived SOA has
the potential for inducing reactive oxygen species (ROS) (Kramer et al., 2016),which are linked
to oxidative stress and inflammation (Reuter et al., 2010; Li et al., 2003). An *in vitro* study that
followed supported the potential for isoprene-SOA to affect the levels of oxidative stress genes
(Lin et al., 2016). In this study we chose to examine the gene expression levels of interleukin-8
(*IL-8*) and cyclooxygenase-2 (*COX-2*), not only for their links to inflammation and oxidative
stress (Kunkel et al., 1991; Uchida, 2008), but because both have been examined in previous
studies using the EAVES for fresh and aged diesel exhaust (Lichtveld et al., 2012). Other studies
on air pollution mixtures have also examined *IL-8* as a biological endpoint due to its involvement
with inflammation (Zavala et al., 2014; Ebersviller et al., 2012a, b; Doyle et al., 2004; Doyle et
al., 2007). We compared the gene expression levels in cells exposed to SOA generated in an
outdoor chamber from photochemical oxidation of isoprene in the presence of NO and acidified
sulfate seed aerosol to cells exposed to a dark control mixture of isoprene, NO, and acidified
sulfate seed aerosol to isolate the effects of the isoprene-derived SOA on the cells using the
EAVES. In addition, we collected SOA onto filters for subsequent resuspension exposure to
ensure that effects observed from EAVES exposures were attributable to particle-phase organic
products.
**2.  Experimental Section**
**2.1 Generation of SOA in the Outdoor Chamber Facility.**   SOA were generated by
photochemically oxidizing a mixture of acidified sulfate seed aerosol, isoprene, and NO injected
into an outdoor smog chamber facility. The outdoor chamber is a 120-m$^3$ triangular cross-section
Teflon chamber located on the roof of the Gillings School of Global Public Health, University of
North Carolina at Chapel Hill. The chamber facility has been described in detail elsewhere by
Lichtveld et al. (2012). The outdoor chamber facility is equipped with sampling lines that allow
direct deposition exposure of cells, online chemical measurements, and filter collection for
offline chemical analysis. Sampling lines run from the underside of the chamber directly to the
chemistry lab below where online measurement instruments and the direct deposition exposure
device are located. Injection ports are also located on the underside of the chamber.

To generate isoprene-derived SOA, the chamber was operated on sunny days, under high

relative humidity, to allow natural sunlight to trigger photochemical reactions. Acidified sulfate
seed aerosols were generated by nebulizing an aqueous solution containing 0.06 M $MgSO_4$ +
0.06 M $H_2SO_4$ into the chamber to a particle concentration of approximately 170 μg m$^{-3}$, which
was allowed to stabilize for 30 min to ensure a well-mixed condition. After stabilization, 3.5
ppmv isoprene (Sigma-Aldrich, 99%) and 200 ppbv NO (AirGas, 1.00%) were injected into the
chamber. Photochemical aging was allowed for approximately one hour to reach the desired
exposure conditions of 30-40 μg m$^{-3}$ growth of isoprene-derived SOA on the pre-existing 170 μg
m$^{-3}$ of acidified sulfate aerosol. This chamber experiment was replicated on three separate sunny
days with temperatures ranging from 24.9°C to 26.8°C with a relative humidity of approximately
70% in the chamber.
**2.2 Control Chamber Experiments.**  As a dark chamber control, to isolate the effect of SOA on
exposed cells, mixtures of isoprene, NO, and 170 μg m$^{-3}$ of acidified sulfate seed aerosol were
injected into the chamber in the dark (after sunset). Conducting the chamber experiments in the
dark ensured no photochemical oxidation of isoprene. The dark control was replicated on three
different nights. Except for the absence of solar radiation (no SOA), all chamber operations and
exposure conditions were similarly maintained.

As an added control to ensure that the device itself and the cell handling had no

significant effect on cell cytotoxicity, cells were exposed in the EAVES to a clean chamber and
compared to unexposed cells kept in an incubator for the same duration as the exposure. The
cytotoxicity results ensured that there is no effect of chamber conditions and device operation on
the cells.
**2.3 Cell Culture.**  Human bronchial epithelial (BEAS-2B) cells were maintained in keratinocyte
growth medium (KGM BulletKit; Lonza), a serum-free keratinocyte basal medium (KBM)
supplemented with 0.004% of bovine pituitary extract and 0.001% of human epidermal growth
factor, insulin, hydrocortisone, and GA-1000 (gentamicin, amphotericin B), and passaged
weekly. Passage number for photochemical exposures and dark control exposures varied
between 52 and 60. Because BEAS-2B are an immortalized line of human bronchial epithelium,
there are limitations with its use such as it being genetically homogeneous, being a single cell
type, and being SV-40 transformed (Reddel et al., 1988). However, BEAS-2B is a stable,
proliferative cell line shown to be useful in airway inflammation studies such as ours (Devlin et
al., 1994).
**2.4 Direct Deposition Exposure.**  In preparation for air-liquid interface exposures, cells were
seeded onto collagen-coated Millicell cell culture inserts (30 mm diameter, 0.4 μm pore size, 4.2
cm$^2$ filter area; Millipore, Cambridge, MA) at a density of 200,000 cells/well 24 hours prior to
exposure. At the time of exposure, cells reached ~80% confluence, confirmed through
microscopy. Immediately before exposure, cell medium was removed from the apical and
basolateral sides of 2 seeded Millicell cell culture inserts. One insert was transferred to a titanium
dish containing 1.5 mL of keratinocyte basal medium (KBM; Lonza), supplying cells with
nutrients from the basolateral side and constant moisture while allowing exposure to be
performed at an air-liquid interface. The other insert was transferred into a 6 well plate with 2
mL of KBM and placed in the incubator as an unexposed control.
Cells were exposed to chamber-generated isoprene SOA using the EAVES located in the
laboratory directly beneath the outdoor chamber (de Bruijne et al., 2009; Lichtveld et al., 2012).
The EAVES, located in an incubator at 37°C, sampled chamber air at 1 L min$^{-1}$. The target
relative humidity (RH) in the chamber during EAVES exposures was approximately 70%.
Exposure time was one hour commencing when target exposure conditions were achieved in the
outdoor chamber for both photochemical and dark control experiments. Detailed description of
the EAVES can be found in de Bruijne et al. (2009).
Following exposure, the cell culture insert was transferred to a 6-well tissue culture plate
containing 2 mL of fresh KBM. The control Millicell was also transferred to 2 mL of fresh
KBM. Nine hours post-exposure, extracellular medium was collected and total RNA was isolated
using Trizol (Life Technologies), consistent with past studies (de Bruijne et al., 2009).
Extracellular medium and the extracted RNA samples were stored at -20°C and -80°C,
respectively, until further analysis. For quality assurance purposes, the RNA concentration and
integrity were assessed using Nanodrop and Bioanalyzer over the period of storage. No changes
were observed under the given storage conditions.
**2.5 Filter Resuspension Exposure.** Chamber particles were collected, concurrently with
EAVES sampling, onto Teflon membrane filters (47 mm diameter, 1.0 µm pore size; Pall Life
Science) for photochemical (light) and dark chamber experiments to be used for chemical
analysis and resuspension exposures. The resuspension experiments served as a control for
possible effects of gaseous components such as ozone ($O_3$) and $NO_x$ present in the direct
deposition experiments; however, prior studies have shown that gaseous components do not
yield cellular responses within the EAVES device (de Bruijne et al., 2009; Ebersviller et al.,
2012a, b). Mass loadings of SOA collected on the filters were calculated from sampling volumes
and average aerosol mass concentrations in the chamber during the sampling period. A density
correction of 1.6 g cm$^{-3}$ (Riedel et al., 2016) and 1.25 g cm$^{-3}$ (Kroll et al., 2006) was applied to
convert the measured volume concentrations to mass concentrations for the acidified sulfate seed
and SOA growth, respectively. The particles collected on Teflon filter membranes for
resuspension cell exposure were extracted by sonication in high-purity methanol (LC/MS
CHROMASOLV, Sigma-Aldrich). Filter samples from multiple experiments were combined and
the combined filter extract was dried under a gentle stream of nitrogen ($N_2$). KBM medium was
then added into the extraction vials to re-dissolve SOA constituents.

In preparation for filter resuspension exposures, cells were seeded in 24-well plates at a

density of $2.5 \times 10^4$ cells/well in 250 μL of KGM 2 days prior to exposure. At the time of
exposure when cells reached ~80% confluence, cells were washed twice with phosphate buffered
saline (PBS) buffer, and then exposed to KBM containing 0.01 and 0.1 mg mL$^{-1}$ isoprene SOA
extract from photochemical experiment and seed particles from dark control experiments.

Following a 9-hour exposure, extracellular medium was collected and total RNA was

isolated using Trizol (Life Technologies) and stored alongside samples from direct deposition
exposures until further analysis.
**2.6 Chemical and Physical Characterization of Exposure**s.  Online and offline techniques
were used to characterize the SOA generated in the chamber. The online techniques measured
the gas-phase species NO, $NO_x$ and $O_3$ and the physical properties of the aerosol continuously
throughout the chamber experiments. Offline techniques measured aerosol-phase species
collected onto Teflon membrane filters (47 mm diameter, 1.0 μm pore size; Pall Life Science)
from photochemical and dark chamber experiments. Filter samples were stored in 20 mL
scintillation vials protected from light at -20°C until analyses.
Real-time aerosol size distributions were measured using a Differential Mobility
Analyzer (DMA, Brechtel Manufacturing Inc.) coupled to a Mixing Condensation Particle
Counter (MCPC, Model 1710, Brechtel Manufacturing Inc.) located in the laboratory directly
underneath the chamber. $O_3$ and $NO_x$ were measured with a ML 9811 series Ozone Photometer
(Teledyne Monitor Labs, Englewood, CO) and ML 9841 series $NO_x$ Analyzer (American
Ecotech, Warren RI), respectively. Data were collected at one-minute intervals using a data
acquisition system (ChartScan/1400) interfaced to a computer. The presence of isoprene in the
chamber was confirmed and quantified using a Varian 3800 gas chromatograph (GC) equipped
with a flame ionization detector (FID).
Chemical characterization of SOA constituents was conducted offline from extracts of
filters collected from chamber experiments by gas chromatography interfaced with an electron
ionization quadrupole mass spectrometer (GC/EI-MS) or by ultra performance liquid
chromatography interfaced with a high-resolution quadrupole time-of-flight mass spectrometer
equipped with electrospray ionization (UPLC/ESI-HR-QTOFMS). Detailed operating conditions
for the GC/EI-MS and UPLC/ESI-HR-QTOFMS analyses as well as detailed filter extraction
protocols have been described previously by Lin et al. (2012). For GC/EI-MS analysis, filter
extracts were dried under a gentle stream of $N_2$ and trimethylsilylated by the addition of 100 μL
of BSTFA + TMCS (99:1 v/v, Supelco) and 50 μL of pyridine (anhydrous, 99.8%, Sigma-
Aldrich) and heated at 70 ºC for 1 h. For UPLC/ESI-HR-QTOFMS analysis, residues of filter
extracts were reconstituted with 150 µL of a 50:50 (v/v) solvent mixture of high-purity water and
methanol.

The isoprene-derived SOA markers: 2-methyltetrols, isomeric 3-methyltetrahydrofurans-

3,4-diols (3-MeTHF-3,4-diols), and 2-methylglyceric acid, synthesized according to the
published procedures (Lin et al., 2013b; Zhang et al., 2012), were available in-house as authentic
standards to quantify the major components of isoprene SOA. 2-Methyltetrol organosulfates,
synthesized as a mixture of tetrabutylammonium salts, were also available as a standard. Purity
was determined to be >99% by [1]H NMR and UPLC/ESI-QTOFMS analysis (Budisulistiorini et
al., 2015b). The $C_5$-alkene triols and IEPOX dimer were quantified using the response factor
obtained for the synthetic 2-methyltetrols.

A representative ambient $PM_{2.5}$ sample collected from the rural southeastern U.S.

(Yorkville, GA) (Lin et al., 2013a) during the summer of 2010 was analyzed in an identical
manner to confirm atmospheric relevance of the chamber-generated SOA constituents.
**2.7 Cytotoxicity Assay.**    Cytotoxicity was assessed through measurement of lactate
dehydrogenase (LDH) released into the extracellular medium from damaged cells using the LDH
cytotoxicity detection kit (Takara). To ensure that the EAVES device itself and operation
procedure had no effect on cytotoxicity, the LDH release from cells exposed to clean chamber air
was measured. LDH release by cells exposed via the EAVES to the photochemically aged (light)
and non-photochemically aged (dark) particles was compared to release from unexposed cells
maintained in the incubator for the same duration. For the resuspension exposures, LDH release
by cells exposed to SOA through resuspended extract of photochemically aged and non-
photochemically aged particles was compared to release by cells maintained in KBM only.
Additionally, LDH release from the light exposures, dark control, and resuspension exposures
was compared to release by positive control cells exposed to 1% Triton X-100 to ensure that cell
death would not affect gene expression results.
**2.8 Gene Expression Analysis.** We chose to measure the levels of the inflammation-related
mRNA in the BEAS-2B cells exposed to isoprene-derived SOA generated in our outdoor
chamber because various particle types are capable of sequestering cytokines (Seagrave, 2008).
Other direct deposition studies have also used mRNA transcripts as a proxy for cytokine
production (Hawley et al., 2014a; Hawley et al., 2014b; Hawley and Volckens, 2013; Volckens
et al., 2009; Lichtveld et al., 2012). Changes in *IL-8* and *COX-2* mRNA levels were measured
using QuantiTect SYBR Green RT-PCR Kit (Qiagen) and QuantiTect Primer Assays for
Hs_ACTB_1_SG (Catalog #QT00095431), Hs_PTGS2_1_SG (Catalog #QT00040586), and
Hs_CXCL8_1_SG (Catalog #QT00000322) for one-step RT-PCR analysis. All mRNA levels
were normalized against β-actin mRNA, which was used as a housekeeping gene. The relative
expression levels (i.e., fold change) of *IL-8* and *COX-2* were calculated using the comparative
cycle threshold ($2^{-\Delta\Delta CT}$) method (Livak and Schmittgen, 2001). For EAVES exposures, changes
in *IL-8* and *COX-2* from isoprene-derived SOA exposed cells were compared to cells exposed to
the dark controls. Similarly, for resuspension exposures changes in *IL-8* and *COX-2* from
isoprene-derived SOA exposed cells were compared to cells exposed to particles collected under
dark conditions.
**2.9 Statistical Analysis.** The software package GraphPad Prism 4 (GraphPad) was used for all
statistical analyses. All data were expressed as mean ± SEM (standard error of means).
Comparisons between data sets for cytotoxicity and gene expression analysis were made using
unpaired *t*-test with Welch's correction. Significance was defined as $p < 0.05$.
**3.   Results and Discussion**
**3.1 Physical and Chemical Characterization of Exposure.**  Figure 1 shows the change in
particle mass concentration and gas ($O_3$, NO, $NO_x$) concentration over time during typical
photochemical and dark control experiments. Under dark control conditions (Fig. 1a) there is no
increase in aerosol mass concentration following isoprene injection. Average total aerosol mass
concentration was $155.0\pm2.69$ µg m$^{-3}$ (1 standard deviation) with no particle mass attributable to
organic material.

In contrast, Fig. 1b shows an increase in aerosol mass concentration after 1 h post

isoprene injection, which can be attributed to the photochemical oxidation of isoprene and
subsequent production and reactive uptake of its oxidation products. The average increase in
aerosol mass concentration attributable to SOA formation for three daylight chamber
experiments conducted on separate days was $44.5\pm5.7$ µg m$^{-3}$. Average total aerosol mass
concentration during particle exposure was $173.1\pm 4.2$ µg m$^{-3}$.

$O_3$ and $NO_x$ concentrations measured during EAVES exposure were approximately 270

ppb and 120 ppb for photochemical experiments. For dark control experiments (e.g., Fig. 1a), the
$O_3$ and $NO_x$ concentrations were approximately 15 ppb and 180 ppb. Previous studies
characterizing the EAVES device show definitively that gas-phase products do not induce cell
response (de Bruijne et al., 2009). However, resuspension exposures were conducted in addition
to EAVES exposure to ensure that biological effects were attributable to only particle-phase
constituents and not gas-phase products such as $O_3$ and $NO_x$.

The chemical composition of aerosol, collected onto filters concurrently with cell

exposure and characterized by GC/EI-MS and UPLC/ESI-HR-QTOFMS, are shown in Fig. 2.
No isoprene-SOA tracers were observed in the filters collected from dark control experiments.
The dominant particle-phase products of the isoprene-SOA collected from photochemical
experiments are derived from the low-NO channel, where IEPOX reactive uptake onto acidic
sulfate aerosol dominates, including 2-methyltetrols, C5-alkene triols, isomeric 3-MeTHF-3,4-
diols, IEPOX-derived dimers, and IEPOX-derived organosulfates. The sum of the IEPOX-
derived SOA constituents quantified by the available standards accounted for ~80% of the
observed SOA mass. The MAE-derived SOA constituents 2-methylglyceric acid and the
organosulfate derivative of MAE, derived from the high-NO channel, accounted for 1.4% of the
observed SOA mass, confirming that particle-phase products generated were predominantly
formed from the reactive uptake of IEPOX onto acidic sulfate aerosols. As demonstrated in
Figure 2, all the same particle-phase products are measured in the $PM_{2.5}$ sample collected in
Yorkville, GA (a typical low-NO region), demonstrating that the composition of the chamber-
generated SOA is atmospherically relevant. Recent SOA tracer measurements from the Southern
Oxidant and Aerosol Study (SOAS) campaign at Look Rock, TN, Centerville, AL, and
Birmingham, AL, also support the atmospheric relevance of IEPOX-derived SOA constituents
that dominate the isoprene SOA mass in summer in the southeastern U.S. (Budisulistiorini et al.,
2015a; Rattanavaraha et al., 2016).
**3.2 Cytotoxicity.** LDH release for cells exposed using the EAVES device is expressed as a fold-
change relative to the unexposed incubator control. For resuspension exposures, LDH release is
expressed as fold-change relative to cells exposed to KBM only. Results shown in Fig. 3a
confirm that there is no effect of chamber conditions and device operation on the cells when
comparing LDH release from cells exposed to a clean air chamber and cells unexposed in an
incubator. Additionally, LDH release from all exposure conditions in EAVES exposed cells (Fig.
3b) and resuspension exposed cells (Fig. 3c) is negligible relative to positive controls exposed to
1% Triton X-100, confirming that the exposure concentration of isoprene-derived SOA utilized
in this study was not cytotoxic. All cytotoxicity results ensured that exposure conditions were not
adversely affecting the cells nor their gene expression.
**3.3 Pro-inflammatory Gene Expression**.  Changes in the mRNA levels of *IL-8* and *COX-2*
from cells exposed to isoprene-derived SOA using the EAVES are shown as fold-changes
relative to dark controls in Fig. 4. This comparison, as well as the results of the resuspension
experiment discussed below, ensure that all effects seen in the cells are attributable to the
isoprene-derived SOA and no other factors. A one-hour exposure to a mass concentration of
approximately 45 µg m$^{-3}$ of organic material was sufficient to significantly alter gene expression
of the inflammatory biomarkers in bronchial epithelial cells. Based on deposition efficiency
characterized by de Bruijne et al. (2009), the estimated dose was 0.29 µg cm$^{-2}$ of total particle
mass with 23% attributable to organic material formed from isoprene photooxidation (0.067 µg
cm$^{-2}$ of SOA).
Changes in the mRNA levels of *IL-8* and *COX-2* from cells exposed to resuspended
isoprene-derived SOA collected from photochemical experiments are shown as fold-changes
relative to cells exposed to resuspended particles from dark control experiments in Fig. 5. At a
low dose of 0.01 mg mL$^{-1}$ of isoprene SOA extract there is no significant increase in *IL-8* and
*COX-2* mRNA expression. The isoprene SOA extract, however, induces a response at a dose of
0.1 mg mL$^{-1}$. The statistically significant increase in mRNA expression from the resuspension
exposure at 0.1 mg mL$^{-1}$ confirms that similar fold changes observed for both *IL-8* and *COX- 2*
from the EAVES exposures are not attributable to gaseous photooxidation products, such as O$_3$,
and support the characterization of the EAVES as a particle exposure device (de Bruijne et al.,

2009).

The similar fold change observed in both the EAVES exposure and resuspension
exposure, in addition to confirming that the biological effects can be attributed to the particle-
phase photochemical products (isoprene-derived SOA), suggests that exposure by resuspension
is appropriate for isoprene-derived SOA and may yield results similar to direct deposition
exposures. Unlike diesel particulate extracts, which agglomerate during resuspension exposures,
isoprene-derived SOA constituents are water-soluble based on reverse-phase LC separations
(Surratt et al., 2006; Lin et al., 2012) and remain well mixed in the cell medium used for
exposure. Therefore, resuspension exposures do not appear to be a limitation for toxicological
assessments of isoprene SOA.
**3.4 Biological Implications**.  The goal of this study was to initially identify potential biological
response associated with exposure to isoprene-derived SOA by using a direct exposure device as
a model that has both atmospheric and physiological relevance. With this model, a dose of 0.067
$\mu$g cm$^{-2}$ of isoprene SOA, induced statistically significant increases in *IL-8* and *COX-2* mRNA
levels in exposed BEAS-2B cells. There are many ways to classify in vitro particle dosimetry
based on the various properties of particles (Paur et al., 2011). For this direct deposition study,
we chose to classify dose as SOA mass deposition per surface area of the exposed cells to mimic
lung deposition. Gangwal et al. (2011) used a multiple-path particle dosimetry (MPPD) model to
estimate that the lung deposition of ultrafine particles ranges from 0.006 to 0.02 $\mu$g cm$^{-2}$ for a 24-
hr exposure to a particle concentration of 0.1 mg m$^{-3}$. Based on this estimate, a dose of 0.067 $\mu$g
cm$^{-2}$ of isoprene SOA in our study can be considered a prolonged exposure over the course of a
week. In fact, most other in vitro studies require dosing cells at a high concentration sometimes
close to a lifetime exposure to obtain a cellular response. Despite this limitation, in vitro
exposures serve as a necessary screening tool for toxicity (Paur et al., 2011).
Our findings are consistent with other studies showing that photochemical oxidation of
similar chemical mixtures increases toxicity in cell culture models and elevates expression of
inflammatory biomarker genes (Lichtveld et al., 2012; Rager et al., 2011). Previous *in vitro*
studies using a gas-phase only exposure system have shown that gas-phase products of isoprene
photooxidation significantly enhance cytotoxicity and *IL-8* expression (Doyle et al., 2004; Doyle
et al., 2007).
By choosing *IL-8* and *COX-2* as our genes of interest, we are able to compare our results
to other studies of known harmful particle exposures. In a similar study using the EAVES,
normal human bronchial epithelial (NHBE) cells exposed to 1.10 µg cm$^{-2}$ diesel particulate
matter showed less than a 2-fold change over controls in both *IL-8* and *COX-2* mRNA
expression (Hawley et al., 2014b). In another study, A549 human lung epithelial cells were
exposed by direct deposition for 1 hour to photochemically-aged diesel exhaust particulates at a
dose of 2.65 µg cm$^{-2}$ from a 1980 Mercedes or a 2006 Volkswagen (Lichtveld et al., 2012).
Exposure to aged Mercedes particulates induced a 4-fold change in IL-8 and ~2-fold change in
*COX-2* mRNA expression, while exposure to aged Volkswagen particulates induced a change of
~1.5-fold in *IL-8* and 2-fold in *COX-2* mRNA expression (Lichtveld et al., 2012). Although the
differences in cell types preclude direct comparisons, the finding of significant increases in *COX-
2* and *IL-8* expression at doses much lower than reported for comparable increases in gene
expression levels induced by photochemically-aged diesel particulates is notable.
*IL-8* and *COX-2* are both linked to inflammation and oxidative stress (Kunkel et al.,
1991; Uchida, 2008). *IL-8* is a potent neutrophil chemotactic factor in the lung and its expression
by various cells plays a crucial role in neutrophil recruitment leading to lung inflammation
(Kunkel et al., 1991). *COX-2* is the inducible form of the cyclooxygenase enzyme, regulated by
cytokines and mitogens, and is responsible for prostaglandin synthesis associated with
inflammation (FitzGerald, 2003). Consistent with the reports that *IL-8* and *COX-2* play important
roles in lung inflammation (Nocker et al., 1996; Li et al., 2013), *in vivo* studies have shown that
isoprene oxidation products cause airflow limitation and sensory irritation in mice (Rohr et al.,
2003). In humans, the role of *IL-8* and *COX-2* in lung inflammation can be associated with
diseases such as chronic obstructive pulmonary disease and asthma (Nocker et al., 1996; Peng et
al., 2008; Fong et al., 2000).

The mechanism by which isoprene-SOA causes elevation of the inflammatory markers

*IL-8* and *COX-2* is not yet fully understood. However, recent work from our laboratory using the
acellular dithiothreitol (DTT) assay demonstrated that isoprene-derived SOA has significant
ROS generation potential (Kramer et al., 2016). High levels of ROS in cells can overwhelm the
antioxidant defense and lead to cellular oxidative stress (Sies, 1991; Bowler and Crapo, 2002; Li
et al., 2003). Following the discovery of the potential importance of isoprene-SOA in generating
ROS, Lin et al. (2016) showed that isoprene-SOA formed from the reactive uptake of epoxides
alters levels of oxidative stress-associated genes, including *COX-2* in human lung cells.
Oxidative stress caused by ROS plays a major role in lung inflammation and the induction of
oxidative stress can lead to *IL-8* expression (Tao et al., 2003; Yan et al., 2015). Specifically,
oxidants can activate the transcription factor NF-κB, which regulates a wide range of
inflammatory genes including *IL-8* and *COX-2* (Barnes and Adcock, 1997; Schreck et al., 1992).
Therefore, isoprene-SOA may cause increases in both *IL-8* and *COX-2* primarily through an
oxidative stress response. Additionally, the relationship between *IL-8* and *COX-2* can also
explain the observed increase in *IL-8* gene expression as the production of *IL*-8 can be stimulated
through a *COX-2* dependent mechanism in airway epithelial cells (Peng et al., 2008).
*In vitro* studies such as this one using a direct deposition model cannot fully elucidate
mechanisms of lung inflammation and potential pathogenesis but serve as a necessary part of
hazard characterization, particularly for a complex air mixture that has not been fully studied
(Hayashi, 2005; Paur et al., 2011). Ozone exposure studies have shown that comparable dose and
effect measurements for *IL-8* and *COX-2* can be found between *in vivo* and *in vitro* exposures
which add promise to extrapolating effects seen *in vitro* to effects *in vivo* (Hatch et al., 2014). In
vivo effects associated with isoprene-SOA exposure *in vitro* cannot be inferred as it is a different
system from ozone, so further *in vitro* studies exploring the health implication of the elevation of
*IL-8* and *COX-2* due specifically to isoprene-SOA exposure are necessary and may in turn justify
further extension to *in vivo* work.
**4. Conclusions**
This study indicates that an atmospherically relevant composition of isoprene-derived
SOA is capable of increasing the expression of *IL-8* and *COX-2* in human bronchial epithelial
cells. The present study is an initial step in a long planned analysis of the biological impacts of
isoprene SOA exposure on lung cells. The SOA were generated as NO levels approached zero,
which represents conditions characteristic of urban locales downwind of rural isoprene sources.
As shown in Fig. 2, the aerosol generated for exposures in this study are chemically similar to
fine aerosol samples collected from the Southeastern U.S., which indicates that the chamber
exposures are representative of exposures that may be encountered by populations in regions
where isoprene emissions interact with anthropogenic pollutants. The same particle-phase
products found in our photochemical experiments have been measured in significant quantities
(accounting on average for 33% of fine organic aerosol mass) in ambient fine organic particles
collected in the Southeastern U.S. (Lin et al., 2013b; Budisulistiorini et al., 2013; Rattanavaraha
et al., 2016; Budisulistiorini et al., 2016) and in other isoprene-rich environments (Hu et al.,
2015). The results of this study show that, because of its abundance, isoprene SOA may be a
public health concern warranting further toxicological investigation through *in vitro* or *in vivo*
work.

**Acknowledgements**
Research described in this article was conducted under contract to the Health Effects Institute
(HEI), an organization jointly funded by the United States Environmental Protection Agency
(EPA) (Assistance Award No. R-82811201), and certain motor vehicle and engine
manufacturers. The contents of this article do not necessarily reflect the views of HEI, or its
sponsors, nor do they necessarily reflect the views and policies of the EPA or motor vehicle and
engine manufacturers. M. A. was supported by a graduate fellowship provided by the National
Science Foundation (DGE-0646083), from the Center for Faculty Excellence, University of
North Carolina at Chapel Hill, and in part by a grant from the National Institute of
Environmental Health Sciences (T32-ES007018).

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

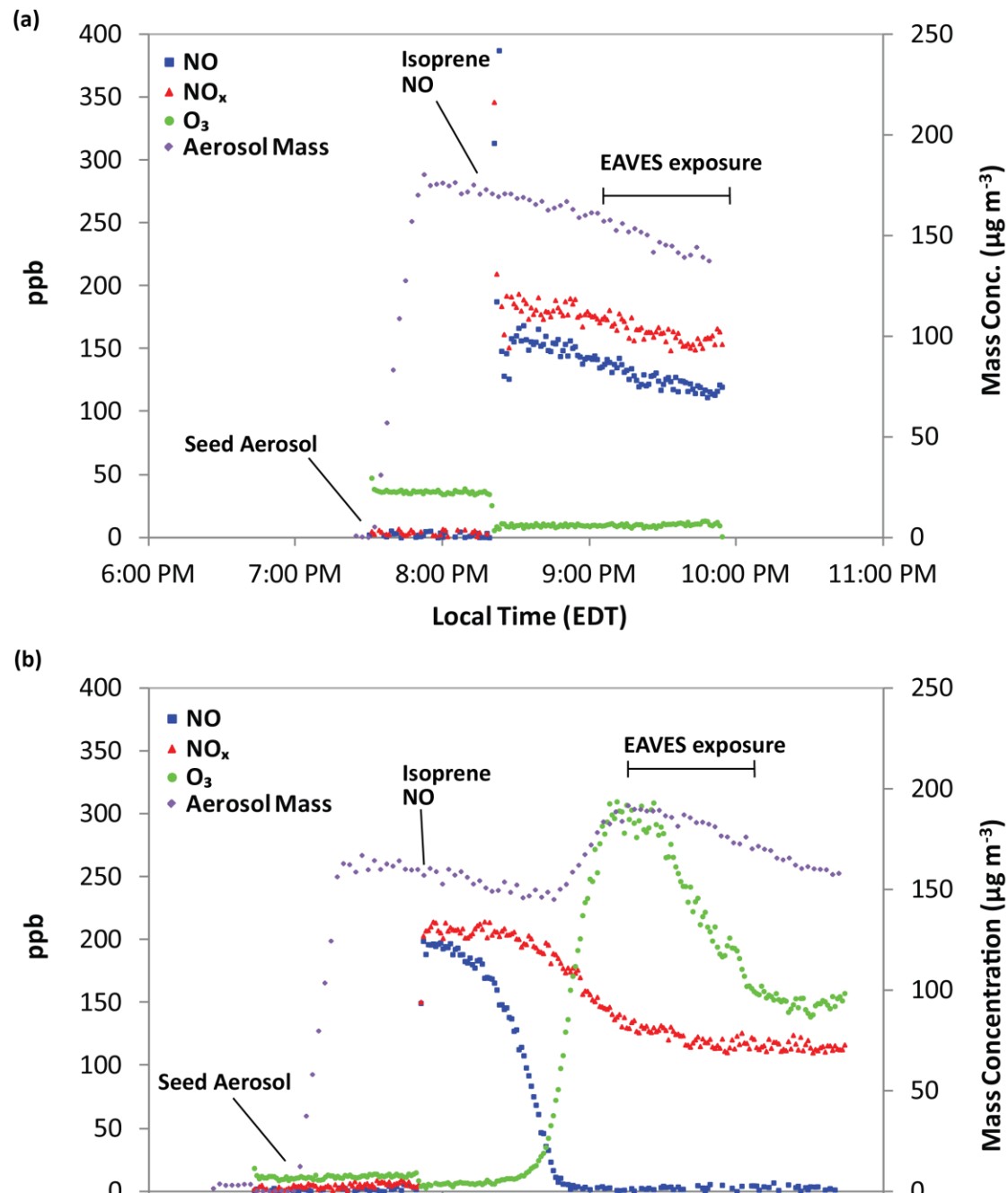

**Figure 1.** Aerosol mass concentration and gas-phase product concentrations over time for (a) dark control chamber experiment and (b) photochemically produced isoprene-derived SOA exposure chamber experiment.


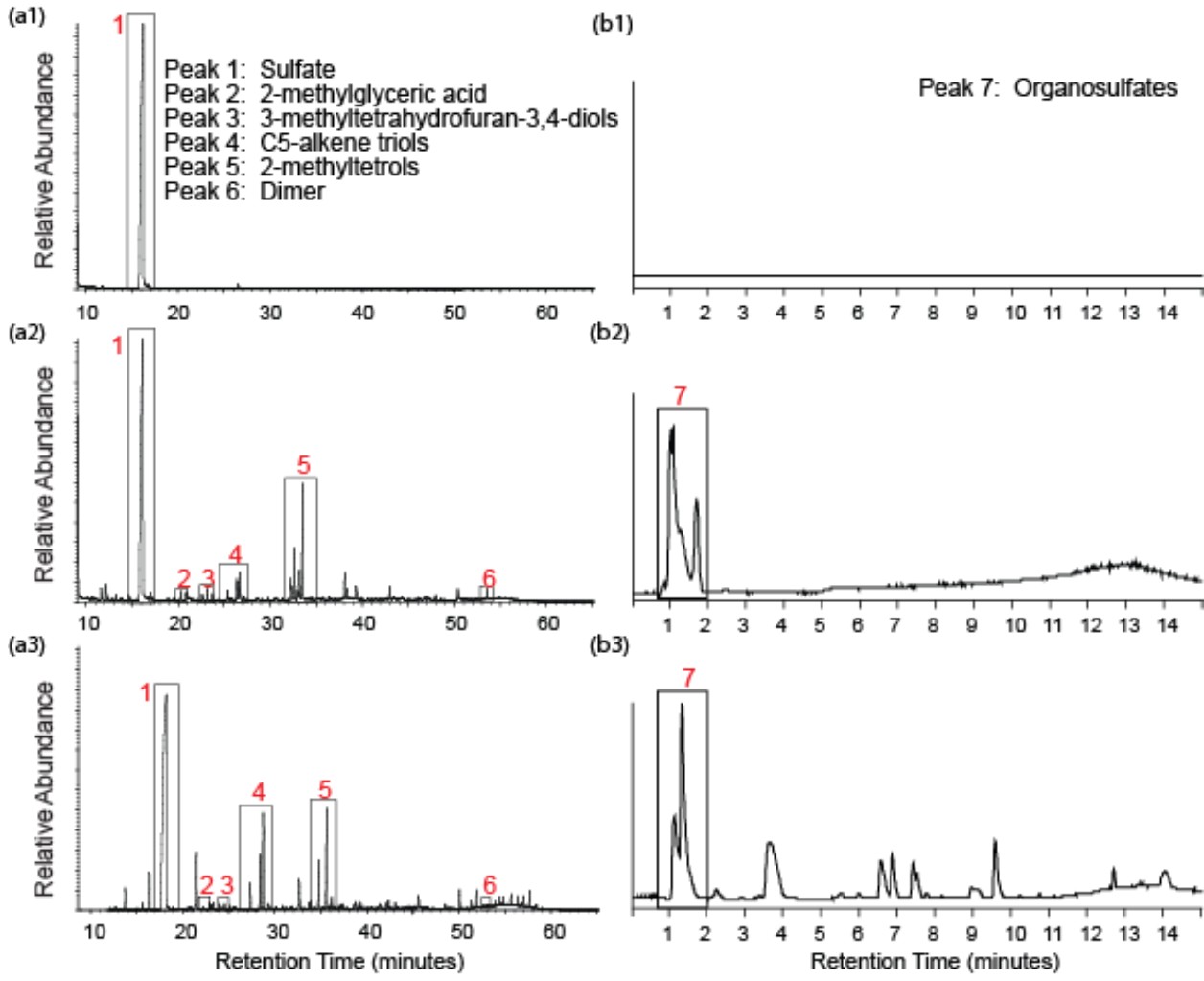


**Figure 2.** (a) GC/EI-MS total ion chromatograms (TICs) and (b) UPLC/ESI-HR-QTOFMS base
peak chromatograms (BPCs) from a (1) dark control chamber experiment, (2) isoprene-derived
SOA exposure chamber experiment, and (3) $PM_{2.5}$ sample collected from Yorkville, GA during
summer 2010.

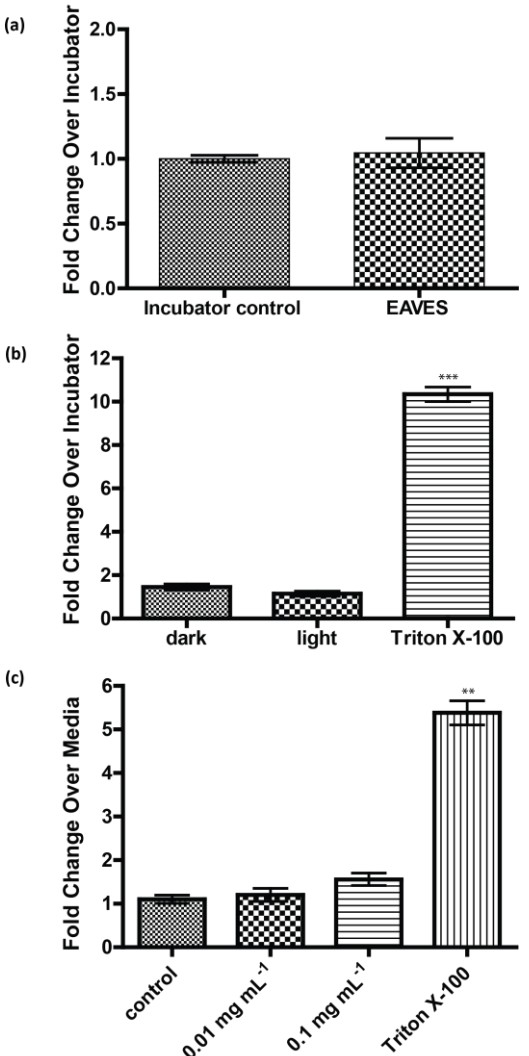


**Figure 3.** LDH release for (a) clean air controls, (b) EAVES exposures, normalized to incubator
control, and (c) resuspension exposures, normalized to KBM only control. **p<0.005 and
***p<0.0005.

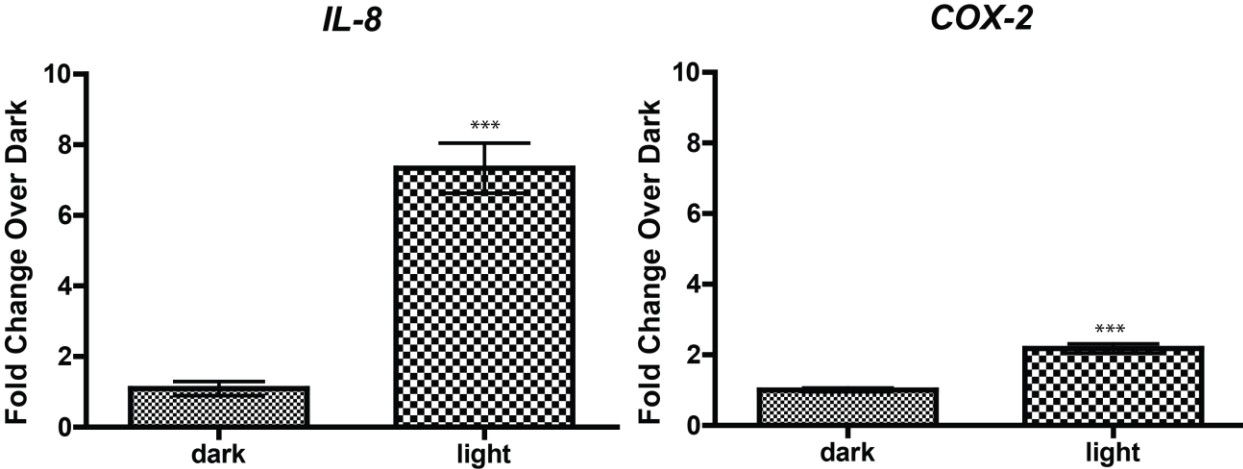

**Figure 4.** *IL-8* and *COX-2* mRNA expression induced by exposure to isoprene-derived SOA
using EAVES device all normalized to dark control experiments and against housekeeping gene,
β-actin. All experiments conducted in triplicate. ***p<0.0005.

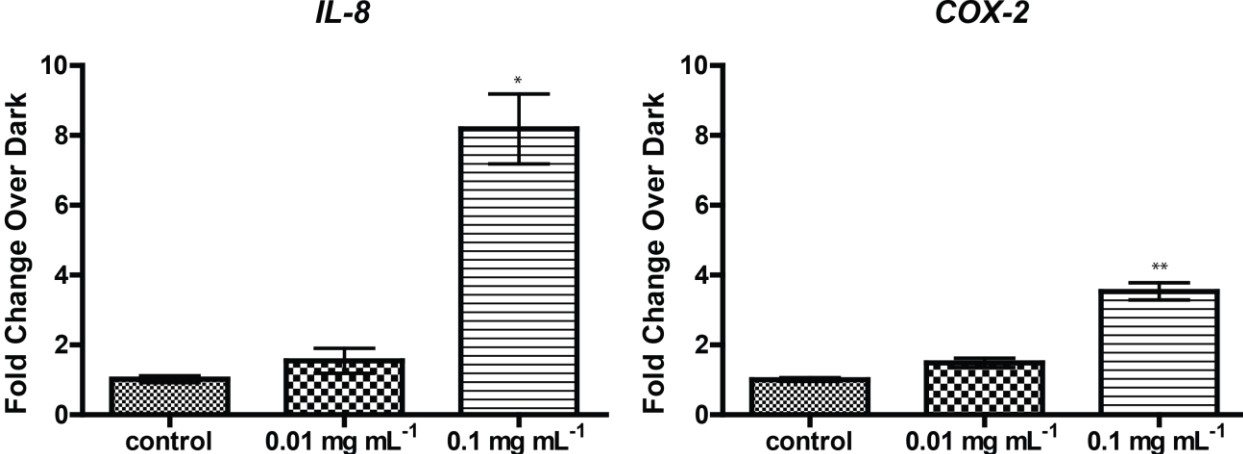


**Figure 5.** *IL-8* and *COX-2* expression induced by exposure to isoprene-derived SOA using
resuspension method all normalized to dark control experiments and against housekeeping gene,
β-actin. All experiments conducted in triplicate. $*p<0.05$ and $**p<0.005$.