# Peer review of "In Vitro Exposure to Isoprene-Derived Secondary Organic Aerosol by Direct Deposition"

_Atmospheric Chemistry and Physics, 2016_

## Referee Comment (RC1) · Anonymous Referee #1 · 8 Jun 2016

The health effects of ambient particulate matter, including SOA components from natural source, is an important scientific concern. Focusing on this issue, this study examined the toxicity of isoprene-derived SOA (generated in an outdoor chamber) on the expression of two inflammation associated genes with an in vitro model of human lung cell line. A novel direct deposition exposure method was applied, and the result was verified with a classical method of resuspended particle exposure. In general, this study was well designed (mainly for the chamber experiment) and has certain scientific significance, therefore it could be considered by the journal of ACP.

A major suggestion is on the discussion section. Obviously, the discussion section was neither in-depth nor penetrating enough, especially for the subsection of "biological implications". In this subsection, it provided only some comparison between this study and others. There is no further discussion on the mechanism between PM exposure and the expression of two inflammation genes, nor any discussion between specific SOA components and gene expression. In addition, why chose mRNA instead of inflammatory factors as the indicator of effects? Increase of gene expression (i.e., mRNA) doesn't always suggest the enhancement of corresponding functional proteins.

The following are some specific comments: Line 82: This abbreviation should be "VOCs". Lines 99-102: The reason for the selection of these two genes was too simple. Suggest the authors to provide some molecular mechanisms between these two genes and oxidative stress and inflammation. Furthermore, this information could also be discussed in the section of results and discussion. Lines 121 and 129: Many factor could influence the photochemical reactions, for example, temperature. What's the temperature (or range) of these sunny days? Lines 134 and 135: Could NO3 radical trigger the formation of SOA at nighttime? Moreover, this statement sounds too assertive, and how about the temperature of the chamber? There must be some difference between nighttime and daytime. Line 151: There is the symbol of "-" between number and unit. Please unify this expression in lines 174 and 201. Line 169: Why choose nine hours as exposure time? Was there any temporal variation during the nine hours? Lin 171: Was there any preliminary experiment to show this storage did not change the extracted mRNA? Line 176: Typo of "resupsension". Lines 197 236: There are two subsection numbers of "2.6". Line 247: Why not measure the inflammatory factors release in the cell culture medium to verify the changes of mRNA? Line 258: Please define the abbreviation of SEM here. Lines 290 to 292: Were there any particular data to support this statement?
* * *

---

## Referee Comment (RC2) · Anonymous Referee #2 · 14 Jun 2016

This paper assesses the toxicity of isoprene SOA through exposure of human lung cells to SOA formed in a chamber. The SOA is deposited directly onto cells and inflammatory biomarkers are monitored. Additional tests with resuspended filter-collected SOA confirms the response is due to particles and not gases formed or originally injected into the chamber (NOx, O3, VOCs). Toxicity is inferred from comparison of the biomarker responses to a seed aerosol (approx. 170 ug/m3 of MgSO4 and H2SO4) to the seed aerosol plus SOA (approx. 170 ug/m3 of acid seed + 30 to 40 ug/m3 isoprene SOA). By essentially noting an increase in the ratio of these biomarkers (SOA+seed/seed) the authors conclude isoprene SOA is toxic to humans. Combined with an earlier paper (Kramer et al., 2016), the authors are asserting that isoprene

[Figure]

SOA is toxic. The results of this paper should be of great interest to the air quality community considering the large implications of what is being proposed; biogenic SOA is toxic, and possibly as toxic as diesel emissions (Kramer et al, 2016). Unfortunately, the results are not highly convincing and I fear that these types of publications generally mislead the community since they leave the impression that biogenic SOA is a health hazard, when really, in this case for example, all they show is that cells responded to very high concentrations of a form of SOA produced in these laboratory experiments. For this reason I do not believe this paper should be published without some major discussion up front qualifying the results.

So, how toxic is isoprene SOA formed under these conditions, is it a health concern? As noted above, a reasonable conclusion from this work is simply for these concentrations, which are much higher than ambient, human lung cells responded, period. If these results could be directly compared to other forms of SOA, than some discussion of relative toxicity could be presented and a context provided. Lack of context is a major flaw and makes the paper results nearly impossible to interpret (see more on this below).

Furthermore, these authors recently published a related manuscript (Lin et al., ES&T letter, 2016), except in Lin et al SOA is formed from reactive uptake of MAE and IEPOX and more genes are measured. In a sense, the materials presented here could have been easily folded into Lin et al to provide more context and would have made a much stronger publication (for both papers). How does one put the findings reported in this work in the context of those reported in Lin et al? Why is this paper not cited in this work?

The following are some major issues.

What type of SOA is being formed? It is not clear chemically, exactly what type of isoprene SOA is being produced in these experiments. Put another way, how does this isoprene compare to what one would be exposed to the ambient environment

(maybe specify specific types of locations). It is not clear how just the presence of certain isoprene tracers observed in both the chamber and at YRK confirm the SOA is identical to ambient (at least identical to what was measured at YRK).

More specifically, it seems that isoprene OA presented in this paper is formed with NO injected into the chamber, with no additional HO2 source. Was isoprene decay measured over time? Under what NOx conditions are most isoprene reacted, and what does the RO2 react with? Self-reaction, with NO, or with HO2? From Figure 1, about half of the SOA is formed where there is NOx. Even after NO is zero, given the large amount of isoprene injected (several ppm), the RO2 + RO2 could be prevalent. It's not clear how "low-NOx" products (RO2+HO2) can be formed in these experiments, and that IEPOX-derived SOA can account for 80% of the SOA formed here. Is an HO2 source added to the chamber? Presumably the SOA in Yorkville is formed under low NOx conditions. More discussed regarding the chamber reactions are needed to justify relevancy to ambient data.

Compare the SOA in these experiments to that presented in their previous paper in Atmos Env (Kramer et al., 2016) where these authors assert that isoprene SOA is as toxic as diesel, based on the DTT assay. It seems the experimental conditions are similar to the manuscript here. However, apparently 2-methylglyceric acid is formed in these experiments (Figure 2 of this manuscript), but not in Kremer et al (Figure 2)? Why? Please provide detailed and specific comparisons on the chemical form of the isoprene formed in these two studies.

Are experiments done under dry or humid conditions?

Issues with Cell Details: The passage numbers used in these experiments seem very high. Please comment on the passage numbers and how determined.

From the results, it doesn't look like the time point is maximized for COX-2. Why was the specific time point used in these experiments they chosen? Is it representative of exposure? Is it to maximize gene expression, etc?

For the filter resuspension exposure, the cells are seeded 2 days prior to exposure and there's no mention of media change. If nutrients are not replenished are the cells highly stressed?

Did the cells exhibit inflammatory responds to the acid seed? Ie, what was the fold increase in the biomarkers for the dark seed experiments to the cells exposed to a completely clean chamber? This might give some sense as to the importance of the fold increase in SOA relative to dark (just seed) aerosol.

Issues with context: The authors state that a dose of 0.067 ug/cm2 to their simulated lung surface is sufficient to induce a response. What is the relevance of this number? Ie, can it be compared to ambient concentrations in any manner, or to say a minimum dose for responses of differing aerosol components in which similar health endpoints were measured? The lung surface area is very large. To have this kind of dose spread throughout the lung would require exposure to an enormous mass of isoprene SOA. The number 0.067 ug/cm2 has little meaning without some context (see more on lack of comparison to other work below).

The final line of the paper illustrates the limitations with lack of context, it states: Taken  together, this study demonstrates that atmospherically relevant compositions of isoprene-derived  SOA can induce adverse effects, suggesting that anthropogenically-derived acidic sulfate aerosol  may drive the generation and toxicity of SOA

This seems too strong a statement, all one may infer from this work is that if you expose lung cells to very high doses of the specific type of isoprene SOA formed in these expts (see questions how atm representative it is), they respond. But cells will respond to many things. Context through relative toxicity could have been provided by doing two identical experiments, but with differing SOA types. Say isoprene vs some aromatic species found in incomplete combustion. There is some discussion near the end of the paper attempting such a comparison, ie comparison to aged diesel exhaust

(Lichtveld et al, 2012), but no definitive answer on the relative toxicity of isoprene SOA can be made because the contrast does not involve identical experiments, (ie, different cell lines were used) making it is difficult to conclude that any observed differences are due solely to the exposure of differing SOA chemical composition. I believe same, applies with Hawley et al, who used primary cells and not a cell line.

The authors further support their observations of inflammatory response due to isoprene SOA by noting they also find that the DTT response for SOA is higher than diesel ((Kramer et al., 2016). What they fail to note is that other analysis, based on ambient data, show a DTT response to isoprene SOA, but it is vastly smaller than the DTT responses to other sources, such as those from incomplete combustion (Verma et al., ES&T, 2015). This again demonstrates the limitation of this work due to lack of context; yes there may be a response to isoprene SOA, but how important is it? These authors may note that the Verma work involved only water-soluble extracts, whereas their experiments involved methanol, and so the difference could be due to non-water soluble isoprene SOA components. But the authors note here that the SOA constituents are "water-soluble (lines 329-330)... and remain well mixed in the cell medium".

Typos: Line 307, should it be Fig 4 and following, Fig 4 should be Fig 5?

---

## Referee Comment (RC3) · Anonymous Referee #3 · 16 Jun 2016

In this paper, the toxicity of isopren -derived secondary organic aerosol (SOA) was examined using the electrostatic aerosol in vitro exposure system (EAVES). The toxicity was evaluated by the lactate dehydrogenase (LDH) assay and also by probing the increase in the inflammatory genes il-8 and cox. Exposures were performed in the light and the dark, for induction of isoprene SOA. The SOA obtained from the EAVES was also compared to PM2.5 collected in Yorkville. Cells maintained in the EAVES system were also compared to cells maintained in regular incubator.

The study is very interesting and provides a new comprehensive approach in understanding the activity of different aerosol components. Overall, the experiments are well described and documented. However, some evidence and logic to explain several is-

sues are still lacking. Major issues: 1. The authors have used 1 hour exposure time. How the setting of 1 hour exposure was chosen? Have different time been measured e.g. longer or shorter than 1 hour? 2. "Photochemical aging was allowed for approximately one hour to reach the desired exposure conditions of 30-40 $\mu$g m-3 growth of isoprene-derived SOA on the pre-existing 170 $\mu$g m-3 of acidified sulfate aerosol" How was this calculation performed? Is this number relevant to real exposure to isoprene SOA? Please also relate to 0.067 ug cm-2.

3. Cytotoxicity measured by LDH is not sufficient for concluding that the isoprene secondary organic aerosol is not toxic. Another assay with a different principal should be performed, such as Hoechst (that interferes with DNA replication and not based on the activity of lactate dehydrogenase enzyme). In addition it would be useful to have an image of the cells before and after exposure?

4. Triton-X 1% raptures the cell's membrane, causing leakage of the inner content of the cells. Therefore, its use as positive control is not be appropriate. It is better to use other cytotoxic agents that are known to cause cell death. 5. What is the biological significance of the increase expression of il-8 and cox genes? Please describe its relevance to a signaling mechanisms that is relevant to isoprene exposure. Minor issues: 6. Materials and methods: 2.3 section should contain the concentration of all the components in the medium, including antibiotics. 7. Section 2.7: add the formation of cDNA using RT (kit, company etc.) 8. Section 2.7: add the primers sequence for both gene tested. 9. There is no reference to Figure 5 in the text. 10. When relating to genes, please use small italics letters (il-8, cox) 11. In figure 2 the a3 graph (on the right panel) the line is in red. This is probably a mistake. If not please add the purpose for the red line in the legend 12. In the graphs indicating fold change, it would be better to write compared to what in the Y axis and not just the legend. Also add information about the normalizing gene in the legend.

---

## Author Comment (AC1) · 24 Aug 2016

**Notes to Editor:**
1. We have highlighted in red all of the changes made for the final submission throughout the main text. These include the changes suggested by the Reviewers.
2. Below we provide a point-by-point response to the Editor and the reviewers.

**Anonymous Referee #1**

The health effects of ambient particulate matter, including SOA components from natural source, is an important scientific concern. Focusing on this issue, this study examined the toxicity of isoprene-derived SOA (generated in an outdoor chamber) on the expression of two inflammation associated genes with an in vitro model of human lung cell line. A novel direct deposition exposure method was applied, and the result was verified with a classical method of resuspended particle exposure. In general, this study was well designed (mainly for the chamber experiment) and has certain scientific significance, therefore it could be considered by the journal of ACP.

A major suggestion is on the discussion section. Obviously, the discussion section was neither in-depth nor penetrating enough, especially for the subsection of "biological implications". In this subsection, it provided only some comparison between this study and others. There is no further discussion on the mechanism between PM exposure and the expression of two inflammation genes, nor any discussion between specific SOA components and gene expression. In addition, why chose mRNA instead of inflammatory factors as the indicator of effects? Increase of gene expression (i.e., mRNA) doesn't always suggest the enhancement of corresponding functional proteins.

Some further discussion about mechanism between exposure and expression of the two inflammation genes has been added on page 17-19, lines 389-423 as follows:

"IL-8 and COX-2 are both linked to inflammation and oxidative stress (Kunkel et al., 1991; Uchida, 2008). IL-8 is a potent neutrophil chemotactic factor in the lung and its expression by various cells plays a crucial role in neutrophil recruitment leading to lung inflammation (Kunkel et al., 1991). COX-2 is the inducible form of the cyclooxygenase enzyme, regulated by cytokines and mitogens, and is responsible for prostaglandin synthesis associated with inflammation (FitzGerald, 2003). Consistent with the reports that IL-8 and COX-2 play important roles in lung inflammation (Li et al., 2013; Nocker et al., 1996), in vivo studies have shown that isoprene oxidation products cause airflow limitation and sensory irritation in mice (Rohr et al., 2003). In humans, the role of IL-8 and COX-2 in lung inflammation can be associated with diseases such as chronic obstructive pulmonary disease and asthma (Fong et al., 2000; Nocker et al., 1996; Peng et al., 2008).

The mechanism by which isoprene-SOA causes elevation of the inflammatory markers IL-8 and COX-2 is not yet fully understood. However, recent work from our laboratory using the acellular dithiothreitol (DTT) assay demonstrated that isoprene-derived SOA have equal or greater ROS generation potential than diesel exhaust PM (Kramer et al., 2016; Rattanavaraha et al., 2011). High levels of ROS in cells can overwhelm the antioxidant defense and lead to cellular oxidative stress (Bowler and Crapo, 2002; Li et al., 2003; Sies, 1991). Following the discovery of the potential importance of isoprene-SOA in generating ROS, Lin et al. (2016) showed that isoprene-SOA formed from the reactive uptake of epoxides alters levels of oxidative stress-associated genes, including COX-2 in human lung cells. Oxidative stress caused by ROS plays a major role in lung inflammation and the induction of oxidative stress can lead to

*IL-8 expression (Tao et al., 2003; Yan et al., 2015). Pathway analysis showed that gene expression of the nuclear factor erythroid 2-related factor 2 (Nrf2) signaling pathway was induced in cells exposed to isoprene-SOA (Lin et al., 2016) which has been reported to alter the expression of IL-8 through mRNA stabilization (Zhang et al., 2005). Therefore, isoprene-SOA may cause increases in both IL-8 and COX-2 primarily through an oxidative stress response. Additionally, the relationship between IL-8 and COX-2 can also explain the observed increase in IL-8 gene expression as the production of IL-8 can be stimulated through a COX-2 dependent mechanism in airway epithelial cells (Peng et al., 2008).*

*In vitro studies such as this one using a direct deposition model cannot fully elucidate mechanisms of lung inflammation and potential pathogenesis but serve as a necessary part of hazard characterization, particularly for a complex air mixture that has not been fully studied (Hayashi, 2005; Paur et al., 2011). Therefore, further in vitro studies exploring the health implication of the elevation of IL-8 and COX-2 due specifically to isoprene-SOA exposure are necessary and may in turn justify further extension to in vivo work."*

The following are some specific comments:

Line 82: This abbreviation should be "VOCs".

This has been corrected in the text on page 4, line 82.

Lines 99-102: The reason for the selection of these two genes was too simple. Suggest the authors to provide some molecular mechanisms between these two genes and oxidative stress and inflammation. Furthermore, this information could also be discussed in the section of results and discussion.

We thank the reviewer for the comment. We believe investigating more genes and molecular mechanisms are the next steps in examining the biological effects of isoprene-SOA. The purpose of this particular study was to serve as an initial step in a long planned analysis of the biological impacts of SOA exposure on lung cells and we anticipate that work on health implications of isoprene-SOA exposure will continue through further *in vitro* and *in vivo* studies. We included statements about the role of *IL-8* and *COX-2* in inflammation and diseases such as chronic obstructive pulmonary disease and asthma but did not want to make overreaching statements about what the elevations of *IL-8* and *COX-2* from our study mean in terms of human health. However, we have included further discussion about the link between the two genes investigated and oxidative stress and inflammation on page 17-19, lines 389-423 as stated in a previous comment.

Additionally, we included citations to other air pollution mixture studies that used *IL-8* as the only gene expression biomarker on page 5, lines 107-110:

*"Other studies on air pollution mixtures have also examined IL-8 as a biological endpoint due to its involvement with inflammation (Doyle et al., 2004; Doyle et al., 2007; Ebersviller et al., 2012a, b; Zavala et al., 2014)."*

Lines 121 and 129: Many factor could influence the photochemical reactions, for example, temperature. What's the temperature (or range) of these sunny days?

The temperature ranged from 24.9°C to 26.8°C at time of isoprene injection on the sunny days. Particles from all experiments were collected onto filters and analyzed to ensure all the same isoprene SOA tracers were measured in all experiments

We have added on page 6, lines 135-137 the temperature range for these photochemical experiments as follows:

*"This chamber experiment was replicated on three separate sunny days with temperatures ranging from 24.9°C to 26.8°C with a relative humidity of approximately 70% in the chamber."*

Lines 134 and 135: Could NO3 radical trigger the formation of SOA at nighttime? Moreover, this statement sounds too assertive, and how about the temperature of the chamber? There must be some difference between nighttime and daytime.

If any $NO_3$ radicals were present, they were likely present at very low mixing ratios as we did not observe any organic particle growth during our nighttime experiments. Additionally, the GC/MS and UPLC/ESI-HR-QTOFMS showed that there were no measurable isoprene-derived SOA tracers (or any other OA species) in the collected filters as shown in Figure 2. We have clarified this in the text on page 13 line 300 as follows:

*"No isoprene-SOA tracers were observed in the filters collected from dark control experiments."*

The reviewer is correct that there are differences in nighttime and daytime temperatures. There were many variables to control when exposing cells using a direct deposition device paired with an atmospheric chamber. To control the temperature issue, the EAVES and incoming air lines were housed in an incubator kept at 37°C as stated in the text on page 8 line 171.

Line 151: There is the symbol of "-" between number and unit. Please unify this expression in lines 174 and 201.

This has been corrected in the text on page 7, lines 160. Also the expressions in line 174 and 201 have been unified and line 201 has been revised to the following on page 9, lines 205-207:

*"Following a 9-hour exposure, extracellular medium was collected and total RNA was isolated using Trizol (Life Technologies) and stored alongside samples from direct deposition exposures until further analysis."*

Line 169: Why choose nine hours as exposure time? Was there any temporal variation during the nine hours?

For the purpose of this study, nine hour post-collection time point was chosen to be consistent with Lichtveld et al. (2012) who used the same exposure device on this outdoor chamber facility. There will be changes to the levels of mRNA expressed at any given point of collection and gene expression does have a time profile. Because of the nature of these chamber exposures, one post-collection time point was chosen and the isoprene-SOA exposures were compared to the dark control exposure for that specific time. However, a time course analysis was conducted using resuspension techniques to show that *COX-2* and *IL-8* are maximized at 9 hours as shown in the graph below.

[Figure]

Line 171: Was there any preliminary experiment to show this storage did not change the extracted mRNA?

We have assessed the integrity of extracted RNA samples after 12 months of storage at -80°C using Nanodrop and Bioanalyzer. We did not observe changes in RNA quality and concentration.

To address this question, we have added this information on page 8, lines 181-183, as follows:

*"For quality assurance purposes, the RNA concentration and integrity were assessed using Nanodrop and Bioanalyzer over the period of storage. No changes were observed under the given storage conditions."*

Line 176: Typo of "resupsension".

This has been corrected in the text on page 8, lines 187.

Lines 197 236: There are two subsection numbers of "2.6".

We have corrected the subsection numbers as the Reviewer pointed out.

Line 247: Why not measure the inflammatory factors release in the cell culture medium to verify the changes of mRNA?

We agree that measuring protein release in the cell culture medium to verify the changes of mRNA would be ideal. However, with the direct deposition exposure method, the particles deposit directly onto the cells and interfere with cytokine measurements as found in Seagraves (2008) so measurements of mRNA was chosen over measurement of inflammatory factors. This information has been added on page 12 line 258-263 as follows:

*"We chose to measure the levels of the inflammation-related mRNA in the BEAS-2B cells exposed to isoprene-derived SOA generated in our outdoor chamber because various particle types are capable of sequestering cytokines (Seagrave, 2008). Other direct deposition studies have also used mRNA transcripts as a proxy for cytokine production (Hawley et al., 2014a; Hawley et al., 2014b; Hawley and Volckens, 2013; Volckens et al., 2009; Lichtveld et al., 2012)."*

To verify if our particles of interest also interfered with cytokine measurements, we first confirmed through an ELISA assay that our cells were capable of releasing IL-8 by spiking them with TNF-α as shown in the left graph below. We then exposed cells using the EAVES to our acidified sulfate seed aerosol and spiked them with TNF-α post-exposure and found that no IL-8 could be measured through the ELISA assay as shown in the right graph below. This information supported our decision to measure mRNA levels as a proxy for cytokine production.

[Figure]

Line 258: Please define the abbreviation of SEM here.

SEM has now been defined: standard error of the mean (SEM) in the text on page 12, line 275.

Lines 290 to 292: Were there any particular data to support this statement?

Filters collected in Yorkville, GA were analyzed using GC/EI-MS and UPLC/ESI-HR-QTOFMS, as shown in Figure 2. Standards were used to identify the isoprene-derived SOA markers. The chamber-generated SOA were collected onto filters and analyzed in the exact same manner and the same pattern of isoprene-derived SOA markers were identified as shown in Figure 2.

We have revised this sentence on page 14, lines 309-312, as follows:

"As demonstrated in Figure 2, all the same particle-phase products are measured in the $PM_{2.5}$ sample collected in Yorkville, GA (a typical low-NO region), demonstrating that the composition of the chamber-generated SOA is atmospherically relevant"

---

## Author Comment (AC2) · 24 Aug 2016

**Notes to Editor:**
1. We have highlighted in red all of the changes made for the final submission throughout the main text. These include the changes suggested by the Reviewers.
2. Below we provide a point-by-point response to the Editor and the reviewers.

**Anonymous Referee #2**

This paper assesses the toxicity of isoprene SOA through exposure of human lung cells to SOA formed in a chamber. The SOA is deposited directly onto cells and inflammatory biomarkers are monitored. Additional tests with resuspended filter-collected SOA confirms the response is due to particles and not gases formed or originally injected into the chamber (NOx, O3, VOCs). Toxicity is inferred from comparison of the biomarker responses to a seed aerosol (approx. 170 ug/m3 of MgSO4 and H2SO4) to the seed aerosol plus SOA (approx. 170 ug/m3 of acid seed + 30 to 40 ug/m3 isoprene SOA). By essentially noting an increase in the ratio of these biomarkers (SOA+seed/seed) the authors conclude isoprene SOA is toxic to humans. Combined with an earlier paper (Kramer et al., 2016), the authors are asserting that isoprene SOA is toxic. The results of this paper should be of great interest to the air quality community considering the large implications of what is being proposed; biogenic SOA is toxic, and possibly as toxic as diesel emissions (Kramer et al, 2016). Unfortunately, the results are not highly convincing and I fear that these types of publications generally mislead the community since they leave the impression that biogenic SOA is a health hazard, when really, in this case for example, all they show is that cells responded to very high concentrations of a form of SOA produced in these laboratory experiments. For this reason I do not believe this paper should be published without some major discussion up front qualifying the results.

Our intention was not for the reader to assume our conclusion was that isoprene-derived SOA is toxic to humans, but rather it induces inflammatory gene expression in the exposed human lung cells and warrants further study. Although the Kramer et al. (2016) study shows that ROS potential (using the DTT assay) of isoprene-derived SOA is similar to some previously assessed diesel particles, this does not necessarily mean these isoprene-derived SOA components are more toxic to humans. To really make this judgment, *in vitro* and *in vivo* studies are needed. We therefore initiated work to expand on the Kramer et al. (2016) study by examining the potential adverse biological effects within human lung cells resulting from isoprene-derived SOA exposure, with a specific focus on inflammatory-related genes examined in past studies (Doyle et al., 2004; Doyle et al., 2007; Hawley et al., 2014b; Lichtveld et al., 2012). Further *in vitro* studies will allow for more exploration of mechanisms of inflammation, whereas chemical-based assays (like DTT) mimic redox reaction potential within organisms that may lead to oxidative stress and eventually inflammation.

To clarify our intention of our study, we have added the following text to the biological implications section (page 19, lines 418-423):

*"In vitro* studies such as this one using a direct deposition model cannot fully elucidate mechanisms of lung inflammation and potential pathogenesis but serve as a necessary part of hazard characterization, particularly for a complex air mixture that has not been fully studied (Hayashi, 2005; Paur et al., 2011). Therefore, further *in vitro* studies exploring the health implication of the elevation of *IL-8* and *COX-2* due

*specifically to isoprene-SOA exposure are necessary and may in turn justify further extension to in vivo work."*

Regarding the high concentrations of isoprene-SOA in the chamber, the high aerosol seed concentration was needed to produce atmospherically relevant compositions of isoprene-SOA. In particular, we selected the conditions of our experiments to mimic compositions of southeastern U.S. aerosol, as shown in Figure 2 and in several recent publications (Rattanvaraha et al., 2016, ACP; Budisulistiorini et al., 2015, ACP). Additionally, a high concentration of particles was needed in the chamber to dose a reasonable amount of particles onto the cells using the direct deposition device.

As stated in the text on page 15, lines 333-335, *"Based on deposition efficiency characterized by de Bruijne et al. (2009), the estimated dose was 0.29 µg cm$^{-2}$ of total particle mass with 23% attributable to organic material formed from isoprene photooxidation (0.067 µg cm$^{-2}$ of SOA)."*

We put this dose into the context of an exposure through the addition of the following text in the discussion on page 16, lines 360-369:

*"There are many ways to classify in vitro particle dosimetry based on the various properties of particles (Paur et al., 2011). For this direct deposition study, we chose to classify dose as SOA mass deposition per surface area of the exposed cells to mimic lung deposition. Gangwal et al. (2011) used a multiple-path particle dosimetry (MPPD) model to estimate that the lung deposition of ultrafine particles ranges from 0.006 to 0.02 µg cm$^{-2}$ for a 24-hr exposure to a particle concentration of 0.1 mg m$^{-3}$. Based on this estimate, a dose of 0.067 µg cm$^{-2}$ of isoprene SOA in our study can be considered a prolonged exposure over the course of a week. In fact, most other in vitro studies, require dosing cells at a high concentration sometimes close to a lifetime exposure to obtain a cellular response. Despite this limitation, in vitro exposures serve as a necessary screening tool for toxicity (Paur et al., 2011)."*

So, how toxic is isoprene SOA formed under these conditions, is it a health concern? As noted above, a reasonable conclusion from this work is simply for these concentrations, which are much higher than ambient, human lung cells responded, period. If these results could be directly compared to other forms of SOA, than some discussion of relative toxicity could be presented and a context provided. Lack of context is a major flaw and makes the paper results nearly impossible to interpret (see more on this below).

The present study is an initial step in long planned analysis of the biological impacts of isoprene SOA exposure on lung cells. More comprehensive studies encompassing expanded gene expression analysis and dose-response relationships will further inform the evaluation of the potential for toxicity. We do not intend to make conclusive statements about isoprene-SOA being a health concern when it is still a new topic of study. In addition to the newly added text in the biological implication as stated in a previous comment, we have added this point in the conclusion sections on page 19 on line 427-428 as follows:

*"The present study is an initial step in a long planned analysis of the biological impacts of isoprene SOA exposure on lung cells"*

Additionally, we have revised the final line in the abstract to more accurately reflect our intentions of the paper as follows:

*"The present study is an attempt to examine the early biological responses of isoprene SOA exposure in human lung cells."*

Furthermore, these authors recently published a related manuscript (Lin et al., ES&T letter, 2016), except in Lin et al SOA is formed from reactive uptake of MAE and IEPOX and more genes are measured. In a sense, the materials presented here could have been easily folded into Lin et al to provide more context and would have made a much stronger publication (for both papers). How does one put the findings reported in this work in the context of those reported in Lin et al? Why is this paper not cited in this work?

Based on Lin et al. (2016), the purpose of this paper was to investigate the effects of photochemically generated isoprene-SOA using our more complex outdoor photochemical chamber. Furthermore, unlike Lin et al. (2016), this paper focuses on utilizing a direct deposition method to better mimic inhalation exposure as stated in the introduction on page 4, lines 87-88, and on page 5, lines 98-99 as follows:

*"The objective of this study is to generate atmospherically relevant isoprene-derived SOA and examine its toxicity through in vitro exposures using a direct deposition device"*

*"Additionally, for a more atmospherically relevant exposure, isoprene-SOA was photochemically generated in an outdoor chamber to mimic its formation in the atmosphere."*

The paper was not cited because at the time of submission, Lin et al. (2016) had not been published. This work is now cited in the text as a potential explanation of the elevations of the two genes studied. This has been added in the biological implications section on page 18, lines 405-413.

*"Following the discovery of the potential importance of isoprene-SOA in generating ROS, Lin et al. (2016) showed that isoprene-SOA formed from the reactive uptake of epoxides alters levels of oxidative stress-associated genes, including COX-2 in human lung cells. Oxidative stress caused by ROS plays a major role in lung inflammation and the induction of oxidative stress can lead to IL-8 expression (Tao et al., 2003; Yan et al., 2015). Pathway analysis showed that gene expression of the nuclear factor erythroid 2-related factor 2 (Nrf2) signaling pathway was induced in cells exposed to isoprene-SOA (Lin et al., 2016) which has been reported to alter the expression of IL-8 through mRNA stabilization (Zhang et al., 2005)."*

Lin et al. (2016) has also now been cited in the introduction on page 4, lines 85-86:

*"…and recently isoprene-SOA formed from the reactive uptake of epoxides has been shown to induce the expression of oxidative stress genes (Lin et al., 2016)."*

and on page 5, lines 102-104

*"An in vitro study that followed supported the potential for isoprene-SOA to affect the levels of oxidative stress genes (Lin et al., 2016)."*

The following are some major issues.

What type of SOA is being formed? It is not clear chemically, exactly what type of isoprene SOA is being produced in these experiments. Put another way, how does this isoprene compare to what one would be exposed to the ambient environment (maybe specify specific types of locations). It is not clear how

just the presence of certain isoprene tracers observed in both the chamber and at YRK confirm the SOA is identical to ambient (at least identical to what was measured at YRK).

As shown in Figure 2, the vast majority of the isoprene-derived SOA tracers measured and quantified using GC/EI-MS and UPLC/ESI-HR-QTOFMS are derived from the low-NO channel, where IEPOX reactive uptake onto acidic sulfate aerosol dominates. Our purpose of using the Yorkville, GA (YRK) sample as an example was to demonstrate that this was the case as YRK is a low-NO region. Recent SOA tracer measurements from the Southern Oxidant and Aerosol Study (SOAS) campaign made by our group at Look Rock, TN, Centerville, AL, and Birmingham, AL, also show that the IEPOX-derived SOA constituents dominate the isoprene SOA mass in summer, even in urban areas like Birmingham, AL (Budisulistiorini et al., 2015b; Rattanavaraha et al., 2016). In addition, we have shown that even in downtown Atlanta, GA, that IEPOX-derived SOA dominates the isoprene SOA mass (Budisulistiorini et al., 2013; Budisulistiorini et al., 2016). We have added this point concerning IEPOX-derived SOA tracers dominating the isoprene SOA mass in ambient $PM_{2.5}$ on page 13-14, lines 298-316, as follows:

*"The chemical composition of aerosol, collected onto filters concurrently with cell exposure and characterized by GC/EI-MS and UPLC/ESI-HR-QTOFMS, are shown in Fig. 2. No isoprene-SOA tracers were observed in the filters collected from dark control experiments. The dominant particle-phase products of the isoprene-SOA collected from photochemical experiments are derived from the low-NO channel, where IEPOX reactive uptake onto acidic sulfate aerosol dominates, including 2-methyltetrols, $C_5$-alkene triols, isomeric 3-MeTHF-3,4-diols, IEPOX-derived dimers, and IEPOX-derived organosulfates. The sum of the IEPOX-derived SOA constituents quantified by the available standards accounted for ~80% of the observed SOA mass. The MAE-derived SOA constituents 2-methylglyceric acid and the organosulfate derivative of MAE, derived from the high-NO channel, accounted for 1.4% of the observed SOA mass, confirming that particle-phase products generated were predominantly formed from the reactive uptake of IEPOX onto acidic sulfate aerosols. As demonstrated in Figure 2, all the same particle-phase products are measured in the $PM_{2.5}$ sample collected in Yorkville, GA (a typical low-NO region), demonstrating that the composition of the chamber-generated SOA is atmospherically relevant. Recent SOA tracer measurements from the Southern Oxidant and Aerosol Study(SOAS) campaign at Look Rock, TN, Centerville, AL, and Birmingham, AL, also support the atmospheric relevance of IEPOX-derived SOA constituents that dominate the isoprene SOA mass in summer in the southeastern U.S. (Budisulistiorini et al., 2015a; Rattanavaraha et al., 2016)."*

More specifically, it seems that isoprene OA presented in this paper is formed with NO injected into the chamber, with no additional HO2 source. Was isoprene decay measured over time? Under what NOx conditions are most isoprene reacted, and what does the RO2 react with? Self-reaction, with NO, or with HO2? From Figure 1, about half of the SOA is formed where there is NOx. Even after NO is zero, given the large amount of isoprene injected (several ppm), the RO2 + RO2 could be prevalent. It's not clear how "low-NOx" products (RO2+HO2) can be formed in these experiments, and that IEPOX-derived SOA can account for 80% of the SOA formed here. Is an HO2 source added to the chamber? Presumably the SOA in Yorkville is formed under low NOx conditions. More discussed regarding the chamber reactions are needed to justify relevancy to ambient data.

Isoprene decay was measured over time to identify its presence in the chamber and whether it reacted. The following is an example of the measured isoprene decay in our photochemical experiments as measured by the GC/FID.

[Figure]

**Isoprene decay during photochemical experiment**

Our goal of using the high isoprene/NO ratio was to create an SOA composition similar to the southeastern U.S. Our chemical results shown in Figure 2 clearly support this, and since we were focusing our cellular exposures to isoprene SOA, we believe we achieved our goal.

To clarify the reviewer's questions, $HO_2$ in the chamber was rapidly formed during the OH-initiated oxidation (i.e., photooxidation) of isoprene. Although we did not add additional sources of OH to the chamber, the photolysis of nitrous acid (HONO) formed at chamber walls provides an intrinsic source of OH radical formation. In addition, the photochemical chamber experiments were conducted at high relative humidity (~70%). The photolysis ozone in the presence of water vapor also provides a source of OH radical.

We believe the majority of $RO_2$ in our chamber experiment reacts with $HO_2$, as demonstrated by our particle phase chemical characterization data. However, we agree with the reviewer that the $RO_2+RO_2$ self reactions could be prevalent in a chamber experiment when the initial isoprene/NO ratio is too high and produces aerosol via nucleation. Based on our aerosol size distribution data, the SOA formation in our chamber experiment (with initial isoprene of 3.5 ppmv) was mainly via condensation without new particle formation.

**3.5 ppmv isoprene+ 200 ppb NO+ 100 µg m⁻³ seed**

When we increased the initial isoprene/NO ratio (increasing the initial isoprene to 5 ppmv), we did observe new particle formation.

[Figure]

**5 ppmv isoprene+ 200 ppb NO+ 100 μg m$^{-3}$ seed**

Thus, we are confident that our chamber-generated isoprene SOA for cell exposure has an atmospherically relevant chemical composition.

Compare the SOA in these experiments to that presented in their previous paper in Atmos Env (Kramer et al., 2016) where these authors assert that isoprene SOA is as toxic as diesel, based on the DTT assay. It seems the experimental conditions are similar to the manuscript here. However, apparently 2-methylglyceric acid is formed in these experiments (Figure 2 of this manuscript), but not in Kramer et al (Figure 2)? Why? Please provide detailed and specific comparisons on the chemical form of the isoprene formed in these two studies.

The SOA analyzed in Kramer et al. (2016) was the same SOA generated for our cell exposures. 2-MG was present in all SOA but was not labeled in Kramer due to its small quantity. However, its peak was identified in our paper to show its presence as an SOA tracer even though it is minimal due to the dominance of IEPOX derived SOA.

Are experiments done under dry or humid conditions?

The isoprene SOA were generated under humid conditions to ensure cell viability for exposure. Relative humidity in the chamber was at least 70% during isoprene injection for photochemical experiments as stated on page 8, line 172, and added on page 6, lines 135-137:

*"This chamber experiment was replicated on three separate sunny days with temperatures ranging from 24.9°C to 26.8°C with a relative humidity of approximately 70% in the chamber"*

Issues with Cell Details: The passage numbers used in these experiments seem very high. Please comment on the passage numbers and how determined.

Records of passage number were kept each time cells were passaged and only passages 52-60 were used for our exposures. Passage numbers in literature can be much higher such as in Wu et al. (2011) which used BEAS-2B cells having passage numbers 70-80.

From the results, it doesn't look like the time point is maximized for COX-2. Why was the specific time point used in these experiments they chosen? Is it representative of exposure? Is it to maximize gene expression, etc?

For the purpose of this study, nine hour post-collection time point was chosen to be consistent with Lichtveld et al. (2012) who used the same exposure device on this outdoor chamber facility. There will be changes to the levels of mRNA expressed at any given point of collection and gene expression does have a time profile. Because of the nature of these chamber exposures, one post-collection time point was chosen and the isoprene-SOA exposures were compared to the dark control exposure for that specific time. However, a time course analysis was conducted using resuspension techniques to show that *COX-2* and *IL-8* are maximized at 9 hours as shown in the graph below.

[Figure]

For the filter resuspension exposure, the cells are seeded 2 days prior to exposure and there's no mention of media change. If nutrients are not replenished are the cells highly stressed?

The BEAS-2B cell protocol involves a change of media every 3-4 days for cell culture. All cells for the exposures and the corresponding controls were handled in the same way.

Did the cells exhibit inflammatory response to the acid seed? Ie, what was the fold increase in the biomarkers for the dark seed experiments to the cells exposed to a completely clean chamber? This might give some sense as to the importance of the fold increase in SOA relative to dark (just seed) aerosol.

The effect of the acidic seed was tested in our resuspension exposures and determined to have minimal effects (no significant differences) on the BEAS-2B cells when compared to media only. The following graphs shows the fold changes of *IL-8* expression of cells exposed to a clean air chamber compared to cells left in an incubator and cells exposed to the acid seed only compared to cells left in an incubator. There were no significant differences in *IL-8* expression based on the EAVES operation and the seed only exposure. Based on this, we determined that the dark only exposures served as the best control to ensure that no other effects such as the particle concentration, equipment handling, and cell handling had effect on the cells.

[Figure]

**Clean air exposure**

IL-8 expression (fold over controls)

Incubator Control | EAVES Exposure

**Seed only exposure**

IL-8 expression (fold over controls)

Incubator Control | EAVES Exposure

| Unpaired t test with Welch's correction | |
|---|---|
| P value | 0.8729 |
| P value summary | ns |
| Significantly different? (P < 0.05) | No |
| One- or two-tailed P value? | Two-tailed |

| Unpaired t test with Welch's correction | |
|---|---|
| P value | 0.8184 |
| P value summary | ns |
| Significantly different? (P < 0.05) | No |
| One- or two-tailed P value? | Two-tailed |

Issues with context: The authors state that a dose of 0.067 ug/cm2 to their simulated lung surface is sufficient to induce a response. What is the relevance of this number? Ie, can it be compared to ambient concentrations in any manner, or to say a minimum dose for responses of differing aerosol components in which similar health endpoints were measured? The lung surface area is very large. To have this kind of dose spread throughout the lung would require exposure to an enormous mass of isoprene SOA. The number 0.067 ug/cm2 has little meaning without some context (see more on lack of comparison to other work below).

In vitro studies require dosing cells at a high concentration sometimes as high as an exposure experienced over a lifetime (Paur et al., 2011). When compared to other similar *in vitro* studies our dose is much lower. We have compared the dose used in our study to those of diesel studies as stated on page 17, lines 377-388 as follows:

*"In a similar study using the EAVES, normal human bronchial epithelial (NHBE) cells exposed to 1.10 µg cm$^{-2}$ diesel particulate matter showed less than a 2-fold change over controls in both IL-8 and COX-2 mRNA expression (Hawley et al., 2014b). In another study, A549 human lung epithelial cells were exposed by direct deposition for 1 hour to photochemically-aged diesel exhaust particulates at a dose of 2.65 µg cm$^{-2}$ from a 1980 Mercedes or a 2006 Volkswagen (Lichtveld et al., 2012). Exposure to aged Mercedes particulates induced a 4-fold change in IL-8 and ~2-fold change in COX-2 mRNA expression, while exposure to aged Volkswagen particulates induced a change of ~1.5-fold in IL-8 and 2-fold in COX-2 mRNA expression (Lichtveld et al., 2012). Although the differences in cell types preclude direct comparisons, the finding of significant increases in COX-2 and IL-8 expression at doses much lower than reported for comparable increases in inflammatory gene expression levels induced by photochemically-aged diesel particulates is notable."*

In addition, to further put the dose in an exposure context, new text has been added on page 16, lines 360-369 as follows:

*"There are many ways to classify in vitro particle dosimetry based on the various properties of particles (Paur et al., 2011). For this direct deposition study, we chose to classify dose as SOA mass deposition per surface area of the exposed cells to mimic lung deposition. Gangwal et al. (2011) used a multiple-path particle dosimetry (MPPD) model to estimate that the lung deposition of ultrafine particles ranges from*

*0.006 to 0.02 µg cm$^{-2}$ for a 24-hr exposure to a particle concentration of 0.1 mg m$^{-3}$. Based on this estimate, a dose of 0.067 µg cm$^{-2}$ of isoprene SOA in our study can be considered a prolonged exposure over the course of a week. In fact, most other in vitro studies require dosing cells at a high concentration sometimes close to a lifetime exposure to obtain a cellular response. Despite this limitation, in vitro exposures serve as a necessary screening tool for toxicity (Paur et al., 2011).”*

The final line of the paper illustrates the limitations with lack of context, it states: Taken together, this study demonstrates that atmospherically relevant compositions of isoprene-derived SOA can induce adverse effects, suggesting that anthropogenically-derived acidic sulfate aerosol may drive the generation and toxicity of SOA

This seems too strong a statement, all one may infer from this work is that if you expose lung cells to very high doses of the specific type of isoprene SOA formed in these expts (see questions how atm representative it is), they respond. But cells will respond to many things. Context through relative toxicity could have been provided by doing two identical experiments, but with differing SOA types. Say isoprene vs some aromatic species found in incomplete combustion. There is some discussion near the end of the paper attempting such a comparison, ie comparison to aged diesel exhaust (Lichtveld et al, 2012), but no definitive answer on the relative toxicity of isoprene SOA can be made because the contrast does not involve identical experiments, (ie, different cell lines were used) making it is difficult to conclude that any observed differences are due solely to the exposure of differing SOA chemical composition. I believe same, applies with Hawley et al, who used primary cells and not a cell line.

*Along with the changes in text as stated above in the biological implication section, the final line of the paper has been changed for further context as follows (page 20 lines 438-440):*

*“The results of this study show that, because of its abundance, isoprene SOA may be a public health concern warranting further toxicological investigation through in vitro or in vivo work”*

*The authors agree that making direct comparisons is difficult when using different cell lines. The choice to use BEAS-2B cells over A549 was made because BEAS-2B is an immortalized non-cancerous cell line, which provides more consistent and representative results for our study design (compared to the cancerous A549 cells or to the primary cells, where responses are subject to interindividual variances). The purpose of this study was not to determine relative toxicity but to identify isoprene-SOA worthy of further in vitro and in vivo studies, as clarified in the abstract with the inclusion of the following statement on lines 46-47.*

*“The present study is an attempt to examine the early biological responses of isoprene SOA exposure in human lung cells”*

The authors further support their observations of inflammatory response due to isoprene SOA by noting they also find that the DTT response for SOA is higher than diesel ((Kramer et al., 2016). What they fail to note is that other analysis, based on ambient data, show a DTT response to isoprene SOA, but it is vastly smaller than the DTT responses to other sources, such as those from incomplete combustion (Verma et al., ES&T, 2015). This again demonstrates the limitation of this work due to lack of context; yes there may be a response to isoprene SOA, but how important is it? These authors may note that the Verma work involved only water-soluble extracts, whereas their experiments involved methanol, and so the difference could be due to non-water soluble isoprene SOA components. But the authors note here

that the SOA constituents are "water-soluble (lines 329-330). . . and remain well mixed in the cell medium".

We would like to point out that in Verma et al. (2015) the isoprene SOA was identified from aerosol mass spectrometry (AMS) measurements through positive matrix factorization (PMF) analysis. The isoprene SOA factor, or more precisely the 82 factor, has been demonstrated to originate from IEPOX SOA (Budisulistiorini et al., 2013). Also, in our recent study (Lin et al., 2016) we show that IEPOX SOA is a weak inducer of cellular oxidative stress gene expression in BEAS-2B cells. Thus, the conclusions from these studies are fairly consistent. However, as shown in Kramer et al. (2016) isoprene SOA has higher DTT activity compared to IEPOX SOA. The difference between Verma et al. (2015) and Kramer et al. (2016) may be due to organic peroxides not measured as part of the isoprene SOA factor (82 factor) (Riva et al., 2016). Therefore, the inflammatory response (i.e., induction of *COX-2* and *IL-8* gene expression) observed in the present study from isoprene SOA exposure could be modulated by oxidative stress. Additional work is required to validate this hypothesis.

We agree with the reviewer that the non-water soluble isoprene SOA components, such as oligomeric species (Lin et al., 2014) could have been much enriched in methanol extracts. We would like to clarify that a majority of isoprene SOA constituents are water-soluble because of their highly oxygenated character, and they appear to be much more hydrophilic compared to diesel particle extracts and remain well mixed in the cell medium during our resuspension exposure processes.

Typos: Line 307, should it be Fig 4 and following, Fig 4 should be Fig 5?

We thank the reviewer for catching this error. This has been corrected in the text.

---

## Author Comment (AC3) · 24 Aug 2016

**Notes to Editor:**
1. We have highlighted in red all of the changes made for the final submission throughout the main text. These include the changes suggested by the Reviewers.
2. Below we provide a point-by-point response to the Editor and the reviewers.

**Anonymous Referee #3**

In this paper, the toxicity of isoprene-derived secondary organic aerosol (SOA) was examined using the electrostatic aerosol in vitro exposure system (EAVES). The toxicity was evaluated by the lactate dehydrogenase (LDH) assay and also by probing the increase in the inflammatory genes il-8 and cox. Exposures were performed in the light and the dark, for induction of isoprene SOA. The SOA obtained from the EAVES was also compared to PM2.5 collected in Yorkville. Cells maintained in the EAVES system were also compared to cells maintained in regular incubator.

The study is very interesting and provides a new comprehensive approach in understanding the activity of different aerosol components. Overall, the experiments are well described and documented. However, some evidence and logic to explain several issues are still lacking.

Major issues: 1. The authors have used 1 hour exposure time. How the setting of 1 hour exposure was chosen? Have different time been measured e.g. longer or shorter than 1 hour?

We were limited by EAVES operating conditions. We chose a 1 hour exposure time to ensure that the cells exposed at the air liquid interface were not stressed due to drying while maximizing deposited dose. However, exposure to the deposited particles would have continued over the 9 hour post-collection period. The nine hour post-collection time point was chosen to be consistent with Lichtveld et al. (2012) who used the same exposure device on this outdoor chamber facility. There will be changes to the levels of mRNA expressed at any given point of collection and gene expression does have a time profile. Because of the nature of the chamber exposures, one post-collection time point was chosen and the isoprene-SOA exposures were compared to the dark control exposure for that specific time. However, a time course analysis was conducted using resuspension techniques to show that *COX-2* and *IL-8* are maximized at 9 hours as shown in the graph below.

[Figure]

2. "Photochemical aging was allowed for approximately one hour to reach the desired exposure conditions of 30-40 µg m-3 growth of isoprene-derived SOA on the pre-existing 170 µg m-3 of acidified sulfate aerosol" How was this calculation performed? Is this number relevant to real exposure to isoprene SOA? Please also relate to 0.067 ug cm-2.

A Differential Mobility Analyzer (DMA, Brechtel Manufacturing Inc.) coupled to a Mixing Condensation Particle Counter (MCPC, Model 1710, Brechtel Manufacturing Inc.) was used to measure the particle mass concentration in the chamber throughout the experiment.

As stated on page 9, lines 192-195: *"A density correction of 1.6 g $cm^{-3}$ (Riedel et al., 2016) and 1.25 g $cm^{-3}$ (Kroll et al., 2006) was applied to convert the measured volume concentrations to mass concentrations for the acidified sulfate seed and SOA growth, respectively".*

The isoprene-SOA growth was measured by taking the difference of the particle mass concentration before isoprene and NO injection and the mass concentration measured once the reaction stabilized and the mass concentration in the chamber peaked.

To put the dose of 0.067 u*g $cm^{-2}$* in context of an exposure, new text has been added on page 16, lines 360-369 as follows:

*"There are many ways to classify in vitro particle dosimetry based on the various properties of particles (Paur et al., 2011). For this direct deposition study, we chose to classify dose as SOA mass deposition per surface area of the exposed cells to mimic lung deposition. Gangwal et al. (2011) used a multiple-path particle dosimetry (MPPD) model to estimate that the lung deposition of ultrafine particles ranges from 0.006 to 0.02 µg $cm^{-2}$ for a 24-hr exposure to a particle concentration of 0.1 mg $m^{-3}$. Based on this estimate, a dose of 0.067 µg $cm^{-2}$ of isoprene SOA in our study can be considered a prolonged exposure over the course of a week. In fact, most other in vitro studies require dosing cells at a high concentration, sometimes close to a lifetime exposure to obtain a cellular response. Despite this limitation, in vitro exposures serve as a necessary screening tool for toxicity (Paur et al., 2011)."*

3. Cytotoxicity measured by LDH is not sufficient for concluding that the isoprene secondary organic aerosol is not toxic. Another assay with a different principal should be performed, such as Hoechst (that interferes with DNA replication and not based on the activity of lactate dehydrogenase enzyme). In addition it would be useful to have an image of the cells before and after exposure?

Cytotoxicity measured through LDH was not an endpoint of interest and was not intended to be used to determine if isoprene-SOA was toxic or non-toxic to cells. Instead, we used LDH as an initial measure of cytotoxicity to ensure that our exposures were not overly toxic to interfere with subsequent gene expression measurements. Additionally, no observed morphological changes and similar RNA concentration and RNA integrity (260/280 and 260/230 values) in all samples, measured using Nanodrop, helped us assess that our exposures did not affect our subsequent gene expression measurements.

No morphological changes were observed after exposure so images were not collected.

4. Triton-X 1% raptures the cell's membrane, causing leakage of the inner content of the cells. Therefore, its use as positive control is not be appropriate. It is better to use other cytotoxic agents that are known to cause cell death.

Triton-X 1% was used to rupture the cell membrane for a positive control as per the LDH protocol from the manufacturer. The LDH is within the cell membrane and Triton-X 1% serves as a positive control by releasing LDH into the supernatant.

5. What is the biological significance of the increase expression of il-8 and cox genes? Please describe its relevance to a signaling mechanisms that is relevant to isoprene exposure.

We included statements about the role of *IL-8* and *COX-2* in inflammation and diseases such as chronic obstructive pulmonary disease and asthma but did not want to make overreaching statements about what the elevations of *IL-8* and *COX-2* from our study mean in terms of human health. The mechanism by which isoprene-SOA increases the expression of IL-8 and COX-2 is not yet understood but some further discussion about mechanism relating exposure to expression of the two inflammation genes has been added on pages 17-19, lines 389-423 as follows:

*"IL-8 and COX-2 are both linked to inflammation and oxidative stress (Kunkel et al., 1991; Uchida, 2008). IL-8 is a potent neutrophil chemotactic factor in the lung and its expression by various cells plays a crucial role in neutrophil recruitment leading to lung inflammation (Kunkel et al., 1991). COX-2 is the inducible form of the cyclooxygenase enzyme, regulated by cytokines and mitogens, and is responsible for prostaglandin synthesis associated with inflammation (FitzGerald, 2003). Consistent with the reports that IL-8 and COX-2 play important roles in lung inflammation (Li et al., 2013; Nocker et al., 1996), in vivo studies have shown that isoprene oxidation products cause airflow limitation and sensory irritation in mice (Rohr et al., 2003). In humans, the role of IL-8 and COX-2 in lung inflammation can be associated with diseases such as chronic obstructive pulmonary disease and asthma (Fong et al., 2000; Nocker et al., 1996; Peng et al., 2008).*

*The mechanism by which isoprene-SOA causes elevation of the inflammatory markers IL-8 and COX-2 is not yet fully understood. However, recent work from our laboratory using the acellular dithiothreitol (DTT) assay demonstrated that isoprene-derived SOA have equal or greater ROS generation potential than diesel exhaust PM (Kramer et al., 2016; Rattanavaraha et al., 2011). High levels of ROS in cells can overwhelm the antioxidant defense and lead to cellular oxidative stress (Bowler and Crapo, 2002; Li et al., 2003; Sies, 1991). Following the discovery of the potential importance of isoprene-SOA in generating ROS, Lin et al. (2016) showed that isoprene-SOA formed from the reactive uptake of epoxides alters levels of oxidative stress-associated genes, including COX-2 in human lung cells. Oxidative stress caused by ROS plays a major role in lung inflammation and the induction of oxidative stress can lead to IL-8 expression (Tao et al., 2003; Yan et al., 2015). Pathway analysis showed that gene expression of the nuclear factor erythroid 2-related factor 2 (Nrf2) signaling pathway was induced in cells exposed to isoprene-SOA (Lin et al., 2016) which has been reported to alter the expression of IL-8 through mRNA stabilization (Zhang et al., 2005). Therefore, isoprene-SOA may cause increases in both IL-8 and COX-2 primarily through an oxidative stress response. Additionally, the relationship between IL-8 and COX-2 can also explain the observed increase in IL-8 gene expression as the production of IL-8 can be stimulated through a COX-2 dependent mechanism in airway epithelial cells (Peng et al., 2008).*

*In vitro studies such as this one using a direct deposition model cannot fully elucidate mechanisms of lung inflammation and potential pathogenesis but serve as a necessary part of hazard characterization, particularly for a complex air mixture that has not been fully studied (Hayashi, 2005; Paur et al., 2011). Therefore, further in vitro studies exploring the health implication of the elevation of*

*IL-8 and COX-2 due specifically to isoprene-SOA exposure are necessary and may in turn justify further extension to in vivo work."*

Minor issues:
6. Materials and methods: 2.3 section should contain the concentration of all the components in the medium, including antibiotics.

The concentrations of all components in the medium have now been reported in the text on page 7, lines 149-153.

*"Human bronchial epithelial (BEAS-2B) cells were maintained in keratinocyte growth medium (KGM BulletKit; Lonza), a serum-free keratinocyte basal medium (KBM) supplemented with 0.004% of bovine pituitary extract and 0.001% of human epidermal growth factor, insulin, hydrocortisone, and GA-1000 (gentamicin, amphotericin B), and passaged weekly"*

Sterile cell culture techniques were employed and, to prevent low level contamination in the cell culture medium, antibiotics were not added.

7. Section 2.7: add the formation of cDNA using RT (kit, company etc.)

We utilized one-step RT-PCR using the QuantiTect SYBR Green RT-PCR Kit which combines the reverse transcription reaction with the PCR reaction. This has been clarified on page 12, lines 263-266 as follows:

*"Changes in IL-8 and COX-2 mRNA levels were measured in BEAS-2B cells exposed to isoprene-derived SOA generated in our outdoor chamber facility using QuantiTect SYBR Green RT-PCR Kit (Qiagen) and QuantiTect Primer Assays for Hs_ACTB_1_SG (Catalog #QT00095431), Hs_PTGS2_1_SG (Catalog #QT00040586), and Hs_CXCL8_1_SG (Catalog #QT00000322) for one-step RT-PCR analysis"*

8. Section 2.7: add the primers sequence for both gene tested.

According to Qiagen technical services, the primer sequences are proprietary and confidential. For additional information, the primers' unique catalog number has been added to the text on page 12, lines 263-266 as follows:

*"Changes in IL-8 and COX-2 mRNA levels were measured in BEAS-2B cells exposed to isoprene-derived SOA generated in our outdoor chamber facility using QuantiTect SYBR Green RT-PCR Kit (Qiagen) and QuantiTect Primer Assays for Hs_ACTB_1_SG (Catalog #QT00095431), Hs_PTGS2_1_SG (Catalog #QT00040586), and Hs_CXCL8_1_SG (Catalog #QT00000322) for one-step RT-PCR analysis"*

9. There is no reference to Figure 5 in the text.

Figure 5 was mistakenly referred to as Figure 4 in the text. This has been corrected on page 15, Lines 337-339:

*"Changes in the mRNA levels of IL-8 and COX-2 from cells exposed to resuspended isoprene-derived SOA collected from photochemical experiments are shown as fold-changes relative to cells exposed to resuspended particles from dark control experiments in Fig. 5"*

10. When relating to genes, please use small italics letters (il-8, cox)

We have followed the guidelines for human gene nomenclature as listed in Wain (2002) by using upper-case letters.

11. In figure 2 the a3 graph (on the right panel) the line is in red. This is probably a mistake. If not please add the purpose for the red line in the legend 12. In the graphs indicating fold change, it would be better to write compared to what in the Y axis and not just the legend. Also add information about the normalizing gene in the legend.

The line color has been changed from red to black. The y-axis for the graphs indicating fold change now includes what the fold change is compared against. The information about the normalizing gene has been added to the legend.

---

## Author Response (AR3)

*The lack of context of this work remains largely unaddressed. The authors have modified the paper slightly, now asserting that this work is more exploratory in nature and that it shows more research in isoprene SOA toxicity is warranted. All three reviewers, in some form or other, have raised the issue questioning how isoprene SOA causes a potentially adverse health response, or put another way, what is the significance of increases in il-8 and cox-2 genes when exposed to isoprene SOA The author's answer to this question is essentially based on their other reported research. The logic put forward in this paper is summarized in lines 400 to 410 (copied below, next paragraph).*

*"The mechanism by which isoprene-SOA causes elevation of the inflammatory markers  IL-8 and COX-2 is not yet fully understood. However, recent work from our laboratory using the acellular dithiothreitol (DTT) assay demonstrated that isoprene-derived SOA have equal or greater ROS generation potential than diesel exhaust PM (Kramer et al., 2016; Rattanavaraha et  al., 2011). High levels of ROS in cells can overwhelm the antioxidant defense and lead to   cellular oxidative stress (Bowler and Crapo, 2002; Li et al., 2003; Sies, 1991). Following the   discovery of the potential importance of isoprene-SOA in generating ROS, Lin et al. (2016)   showed that isoprene-SOA formed from the reactive uptake of epoxides alters levels of oxidative   stress-associated genes, including COX-2 in human lung cells. Oxidative stress caused by ROS   plays a major role in lung inflammation and the induction of oxidative stress can lead to IL-8 expression (Tao et al., 2003; Yan et al., 2015)."*

*The logic is: Diesel is a known toxic aerosol component (implicitly assumed), and ROS (as measured by the DTT assay) is linked to oxidative stress, which is linked to various adverse health endpoints. Since isoprene SOA has equal of higher DTT per mass activities than diesel, and DTT assay is a measure of aerosol oxidative potential, isoprene may be toxic through an oxidative stress route. The problem with this logic is that it all hinges on isoprene SOA being more toxic than diesel, as reported by (Kramer et al., 2016; Rattanavaraha et al., 2011). These papers indeed show this, but the comparison in these two papers is flawed because, for some reason, the diesel DTT per mass reported in Kramer et al., and Rattanavaraha et al, (note, that Rattanavaraha et al appears to simply reuses Kramer et al data, so it all comes down to Kramer) is vastly lower (by about a factor of 10) than what others have reported in both chamber and ambient studies [Bates et al., 2015; Charrier et al., 2015; McWhinney et al., 2013]. (Note, more papers on ambient DTT per mass activities in locations highly impacted by vehicle emissions also exist, but I have not included them here).*

*The response of the authors to this issue raised in the first review was that Bates et al isoprene DTT per mass was based on AMS PMF data and so the isoprene SOA was a different form than the isoprene SOA in this experiment. But that is not the issue. Bates also gives low DTT per mass activities for Other OA (ie, expected to be mainly biogenic SOA). The issue is not DTT per mass for isoprene, there is actually some agreement on this, the issue is why is Rattanavaraha DTT per mass for diesel so low relative to other studies? By only citing their own work and ignoring other published data the authors have made a substantial flaw in reasoning and also give a biased interpretation. The line, "our laboratory using the acellular dithiothreitol (DTT) assay demonstrated that isoprene-derived SOA have equal or greater ROS generation potential than diesel exhaust PM." is factually correct since the citation is limited to their work, but is known to be much more complicated than they knowledge. It seems to be defined narrowly here simply to support the premise of this paper. This issue must be rectified before publication.*

**Response to Reviewer 3 comments:**

Our intention was not to use the previous DTT assay work to frame the entirety of our work. We cited the work because it is one of the few that has been done on the isoprene-SOA system. Because the DTT assay is an acellular assay it is not an effective way to assess or compare biological effects of isoprene-SOA and diesel exhaust. Instead, we highlight the work of Lichtveld et al. (2012) and Hawley et al. (2014) to better put our results into context of the highly studied system of diesel exhaust and draw comparisons between the effect on *IL-8* and *COX-2* from isoprene-SOA and diesel exhaust as stated in line 378-390.

Despite the use of a different cell line, these studies are more comparable to our study because they are also *in vitro* studies utilizing a direct deposition exposure method to measure the changes in gene expression unlike the acellular chemical based DTT assay measuring redox potential. However, in response to the reviewer's comment, we have revised the statement in lines 402-405 to read:

*"The mechanism by which isoprene-SOA causes elevation of the inflammatory markers IL-8 and COX-2 is not yet fully understood. However, recent work from our laboratory using the acellular dithiothreitol (DTT) assay demonstrated that isoprene-derived SOA has significant ROS generation potential (Kramer et al., 2016)."*

Although the DTT assay in not a focus of this paper, in this response we try to address the reviewer's concerns about the discrepancy in the comparison of ROS generating potential of isoprene-SOA to diesel exhaust. First, Rattanavaraha et al. (2011) does not reuse data from Kramer et al. (2016) but graphical data from Rattanavaraha et al. (2011) is presented in Kramer et al. (2016). Kramer et al. (2016) found that isoprene-SOA show equal or greater ROS generation potential than reported in the study on diesel exhaust conducted by Rattanavaraha et al. (2011) for nighttime diesel with and without ozone and daytime diesel without urban mix. The numbers reported for Rattanavaraha et al. (2011) is likely lower in comparison to the other studies cited by the reviewer because EDTA was used in the DTT assay to remove any interference from transition metals in order to focus specifically on the effects of different organic systems. In Charrier et al. (2015), they conclude that copper and manganese largely account for all of the DTT response from nighttime sources and "daytime sources are also generally dominated by metals, but these samples also have a large (up to 50%) contribution from unknown compounds." In McWhinney et al. (2013) they state that "the redox activity of DEP is not extractable by moderately polar (methanol) and nonpolar (dicholoromethane) organic solvents." They report "only 2-11% of the redox activity was in the water-soluble potion, while the remainder occurred at the black carbon surface." There may have been lower DTT activity reported in Rattanavaraha et al. (2011) for nighttime diesel particles because they utilized a methanol extraction to compare soluble organic components, but found that oxidized PAH products enhaced DTT acivity in daytime diesel samples. As stated in the previous response, the DTT assay in (Bates et al., 2015) is not comparable to Kramer et al. (2016) or Rattanavaraha et al. (2011).

Despite the DTT activity of isoprene-SOA being lower in comparison to the DTT activity measured due to transition metals in ambient samples, the abundance of isoprene-SOA in the atmosphere is not insignificant and its ROS generation potential cannot be discounted. Regardless, our suggested reasoning for increased expressions of *IL-8* and *COX-2* via an oxidative stress pathway hinges more on the finding by the cited Lin et al. (2016), showing altered expression of oxidative stress genes via an *in vitro* study, than Kramer et al. (2016).

In regards to the reviewers comment that "*All three reviewers, in some form or other, have raised the issue questioning how isoprene SOA causes a potentially adverse health response, or put*

*another way, what is the significance of increases in il-8 and cox-2 genes when exposed to isoprene SOA",* we had addressed this issue in our previous edit by including the following (line 391-417):

*"IL-8 and COX-2 are both linked to inflammation and oxidative stress (Kunkel et al., 1991; Uchida, 2008). IL-8 is a potent neutrophil chemotactic factor in the lung and its expression by various cells plays a crucial role in neutrophil recruitment leading to lung inflammation (Kunkel et al., 1991). COX-2 is the inducible form of the cyclooxygenase enzyme, regulated by cytokines and mitogens, and is responsible for prostaglandin synthesis associated with inflammation (FitzGerald, 2003). Consistent with the reports that IL-8 and COX-2 play important roles in lung inflammation (Nocker et al., 1996; Li et al., 2013), in vivo studies have shown that isoprene oxidation products cause airflow limitation and sensory irritation in mice (Rohr et al., 2003). In humans, the role of IL-8 and COX-2 in lung inflammation can be associated with diseases such as chronic obstructive pulmonary disease and asthma (Nocker et al., 1996; Peng et al., 2008; Fong et al., 2000)."*

*The mechanism by which isoprene-SOA causes elevation of the inflammatory markers IL-8 and COX-2 is not yet fully understood. However, recent work from our laboratory using the acellular dithiothreitol (DTT) assay demonstrated that isoprene-derived SOA has significant ROS generation potential (Kramer et al., 2016). High levels of ROS in cells can overwhelm the antioxidant defense and lead to cellular oxidative stress (Sies, 1991; Bowler and Crapo, 2002; Li et al., 2003). Following the discovery of the potential importance of isoprene-SOA in generating ROS, Lin et al. (2016) showed that isoprene-SOA formed from the reactive uptake of epoxides alters levels of oxidative stress-associated genes, including COX-2 in human lung cells. Oxidative stress caused by ROS plays a major role in lung inflammation and the induction of oxidative stress can lead to IL-8 expression (Tao et al., 2003; Yan et al., 2015). Specifically, oxidants can activate the transcription factor NF-κB, which regulates a wide range of inflammatory genes including IL-8 and COX-2 (Barnes and Adcock, 1997; Schreck et al., 1992) Therefore, isoprene-SOA may cause increases in IL-8 and COX-2 primarily through an oxidative stress response. Additionally, the relationship between IL-8 and COX-2 can also explain the observed increase in IL-8 gene expression as its production can be stimulated through a COX-2 dependent mechanism in airway epithelial cells (Peng et al., 2008)."*

Because the gene expression of *COX-2* and *IL-8* can be regulated at various levels and have varying biological effects, we remain cautious about making any conclusive statements about the mechanism in which both were elevated and the resulting health effects from the elevations based on this study alone. However, to provide even further evidence about the importance of *in vitro* work (separate from acellular techniques such as DTT) we have added a citation from Hatch et al. (2014) on page 19 line 418-427 as follows:

*"In vitro studies such as this one using a direct deposition model cannot fully elucidate mechanisms of lung inflammation and potential pathogenesis but serve as a necessary part of hazard characterization, particularly for a complex air mixture that has not been fully studied (Hayashi, 2005; Paur et al., 2011). Ozone exposure studies have shown that comparable dose and effect measurements for IL-8 and COX-2 can be found between in vivo and in vitro exposures which add promise to extrapolating effects seen in vitro to effects in vivo (Hatch et al., 2014). In vivo effects associated with isoprene-SOA exposure in vitro cannot be inferred as it is a different system from ozone, so further in vitro studies exploring the health implication of the elevation of IL-8 and COX-2 due specifically to isoprene-SOA exposure are necessary and may in turn justify further extension to in vivo work."*

[revised manuscript text omitted]